# Homopolymer self-assembly of poly(propylene sulfone) hydrogels via dynamic noncovalent sulfone–sulfone bonding

Fanfan Du[1,2], Baofu Qiao [3], Trung Dac Nguyen[4], Michael P. Vincent[1], Sharan Bobbala[1], Sijia Yi[1], Chamille Lescott[3], Vinayak P. Dravid [3], Monica Olvera de la Cruz [3,5,6,7] & Evan Alexander Scott [1,2,6,7,8 ✉]

Natural biomolecules such as peptides and DNA can dynamically self-organize into diverse hierarchical structures. Mimicry of this homopolymer self-assembly using synthetic systems has remained limited but would be advantageous for the design of adaptive bio/nanomaterials. Here, we report both experiments and simulations on the dynamic network self-assembly and subsequent collapse of the synthetic homopolymer poly(propylene sulfone). The assembly is directed by dynamic noncovalent sulfone–sulfone bonds that are susceptible to solvent polarity. The hydration history, specified by the stepwise increase in water ratio within lower polarity water-miscible solvents like dimethylsulfoxide, controls the homopolymer assembly into crystalline frameworks or uniform nanostructured hydrogels of spherical, vesicular, or cylindrical morphologies. These electrostatic hydrogels have a high affinity for a wide range of organic solutes, achieving >95% encapsulation efficiency for hydrophilic small molecules and biologics. This system validates sulfone–sulfone bonding for dynamic self-assembly, presenting a robust platform for controllable gelation, nanofabrication, and molecular encapsulation.

[1] Department of Biomedical Engineering, Northwestern University, Evanston, IL 60208, USA. [2] Simpson Querrey Institute, Northwestern University, Chicago, IL 60611, USA. [3] Department of Materials Science and Engineering, Northwestern University, Evanston, IL 60208, USA. [4] Department of Chemical and Biological Engineering, Northwestern University, Evanston, IL 60208, USA. [5] Department of Chemistry, Northwestern University, Evanston, IL 60208, USA. [6] Chemistry of Life Processes Institute, Northwestern University, Evanston, IL 60208, USA. [7] Interdisciplinary Biological Sciences Program, Northwestern University, Evanston, IL 60208, USA. [8] Robert H. Lurie Comprehensive Cancer Center, Northwestern University, Chicago, IL 60611, USA. ✉email: evan.scott@northwestern.edu

Self-assembly is ubiquitous in biological systems and underlies the formation of complex structures from simple components. Compared to the capabilities of natural polymers, such as polypeptides[1] and DNA[2], for dynamic assembly and reorganization via noncovalent interactions, synthetic polymer-based systems are relatively primitive[3]. Such controlled aqueous self-assembly of single-component supramolecular systems has remained a challenge for synthetic polymers, which typically require amphiphilicity or direct incorporation of naturally occurring peptide/nucleic acid monomers or derivatives thereof for controlled aggregation in aqueous systems. The most common synthetic self-assembling systems employ amphiphilic copolymers that are limited to separate hydrophobic and hydrophilic segments, which do not allow the complex dynamic self-assembly behavior observed in nature while also maintaining aggregate stability required for a broad range of applications. In terms of nanofabrication in aqueous solution, the focus has been placed on the design of amphiphilic block[4] or random[5] copolymers wherein the adjustment of the volume fraction of the hydrophobic/hydrophilic components permits fabrication of various nanostructures including spheres, cylinders, and vesicles. The multicomponent nature and dauntingly wide range of chemical compositions that have been developed for these self-assembling systems can present difficulties during chemical synthesis[6] and practical application[7], often requiring expertise in synthetic polymer chemistry and delaying translation of useful technologies.

Noncovalent weak/reversible interactions such as $\pi$–$\pi$ stacking[8], hydrogen bonding[9,10] and certain metal–ligand coordination bonds[11] provide strategies for the construction of structurally dynamic architectures. Here, we report on a single-component homopolymer system that assembles through dynamic noncovalent sulfone–sulfone bonding. Crystalline frameworks and nanoscale hydrogels of spherical, vesicular, and cylindrical morphology are controllably assembled from solely a poly(propylene sulfone) (PPSU) homopolymer when transitioning from dimethylsulfoxide (DMSO) solution to the aqueous system. Semi-flexible PPSU chains form electrostatically bound networks that reorganize dynamically as interactions among sulfone repeat units increase. This system mimics the dynamic self-assembly of proteins[12] and DNA[13], allowing the design and fabrication of diverse superstructures capable of highly efficient molecular encapsulation.

## Results

### Synthesis and characterization of PPSU homopolymer.
PPSU can be synthesized from the complete oxidation of poly(propylene sulfide) (PPS)[14,15], which is known for its hydrophobic–hydrophilic transition upon oxidation[16–18]. However, complete oxidation of PPS had not been previously achieved, since standard oxidation of PPS generates random copolymers of poly(propylene sulfoxide)-co-poly(propylene sulfone)[19]. We successfully synthesized $PPSU_{20}$ (20 sulfones, Supplementary Fig. 1) using an atypically high concentration of $H_2O_2$. Highly concentrated $H_2O_2$ was generated in situ upon vacuum evaporation during the oxidizing reaction. The resulting $PPSU_{20}$ is a crystalline solid (Supplementary Fig. 2), and solubility tests in water and common organic solvents (Supplementary Table 1) revealed that only DMSO can effectively break $PPSU_{20}$ crystals into a clear solution. By following the phase transition for a DMSO solution of $PPSU_{20}$ during vacuum evaporation, we found that the system remained homogeneous until drying, suggesting an ultra-high solubility of $PPSU_{20}$ in anhydrous DMSO.

### Structural features of PPSU.
Given that the chemical structure of PPSU (Fig. 1a) is characterized by a positively charged backbone with negatively charged pendent oxygen atoms[20], electrostatic repulsion among oxygen atoms[21] was expected to result in weak inter- and intra-chain associations. We anticipated that the weak intra-chain associations of PPSU in DMSO would be accompanied by high chain stiffness. In order to establish that the structural features of PPSU can give rise to chain expansion and solubilization, we carried out all-atom explicit solvent molecular dynamics (AAMD) simulations for $PPSU_{20}$ in DMSO (simulation parameters are given in Supplementary Table 2 and snapshots of initial and final conformations are shown in Supplementary Fig. 3). A typical snapshot is depicted in Fig. 1b, showing that the initially coiled chains turned into roughly extended conformations (Supplementary Movie 1). The calculated persistence length ($L_p$, Supplementary Table 3) confirmed that the polymer is a semi-flexible chain in DMSO with an average $L_p$ of $9.2 \pm 0.7$ sulfones. The simulations also suggest that inter-chain associations are not completely inhibited in DMSO as indicated by the sulfone–sulfone complementary. Complementarity-driven bundling of $PPSU_{20}$ chains would lead to a distorted two-dimensional (2D) structure with enrichment of oxygen atoms on the surface (Fig. 1c) due to the zigzag trans-planar arrangement of sulfones between two parallel but slightly twisted chains (Supplementary Movie 2). We, therefore, inferred that the inter-chain associations are reversible in DMSO. This self-limited growth is confirmed by both a cluster formation analysis (Supplementary Fig. 4) and the DMSO simulations of more $PPSU_{20}$ chains at a higher concentration (Supplementary Fig. 5). Furthermore, we demonstrated negligible dipolar attractive interactions (Supplementary Fig. 6) for the sulfone–sulfone pairs (Fig. 1d), which are approximately comparable to those of sulfone-DMSO, supporting the ultra-high solubility of $PPSU_{20}$ in DMSO.

### PPSU promotes the formation of sulfone-bond networks in water.
We further investigated the solubility of $PPSU_{20}$ by mixing its DMSO solution with an excess amount of other solvents (Supplementary Table 1). Interestingly, aggregates were visible in highly polar solvents including water and aprotic acetonitrile. In contrast, solvents with lower polarity than DMSO did not induce aggregation. These observations indicate that the highly polar PPSU polymer is not soluble in water as widely predicted[14]. Characterized by a high dipole moment[22] and partially charged groups, the sulfones show a formal similarity to zwitterions, and it seemed possible that PPSU, like zwitterionic polymers[23], may have the ability to form physically cross-linked networks. Although the formation of such sulfone–sulfone bonded networks would be inhibited by electrostatic repulsion among oxygen atoms, we anticipated PPSU to overcome this limitation when exposed to highly polar mediums such as water. AAMD simulations of $PPSU_{20}$ in water provided insight into the mechanism and kinetics of aggregate formation. We observed in the simulations that the initially extended $PPSU_{20}$ chains collapsed within 1 ns of the simulations (Supplementary Movie 3), followed by chain folding and association into three-dimensional (3D) superstructures (Fig. 1e). A significant decrease in both $L_p$ and end-to-end distance for $PPSU_{20}$ chains was discovered in water versus in DMSO (Supplementary Table 3). This conformational transition is consistent with the aggregates expected to form upon hydration of $PPSU_{20}$ chains under conditions of weak electrostatic repulsion. Water molecules generally aid supramolecular assembly by means of hydrogen bonds[24]. However, the $PPSU_{20}$/water AAMD simulations revealed hydrogen bonds to not be a significant driver of aggregation, as water occurred exclusively at the surfaces of the 3D superstructures. Instead of water bridges, only dislocation of oxygen atoms on PPSU between layers were visible in the 3D superstructures (Supplementary Movies 4). Neighboring water molecules were predicted to be located on surfaces but not inside of PPSU aggregates (Fig. 1e). The 3D superstructures are stabilized by strong dipolar attractive interactions (Fig. 1d), suggesting an electrostatic nature of the sulfone–sulfone

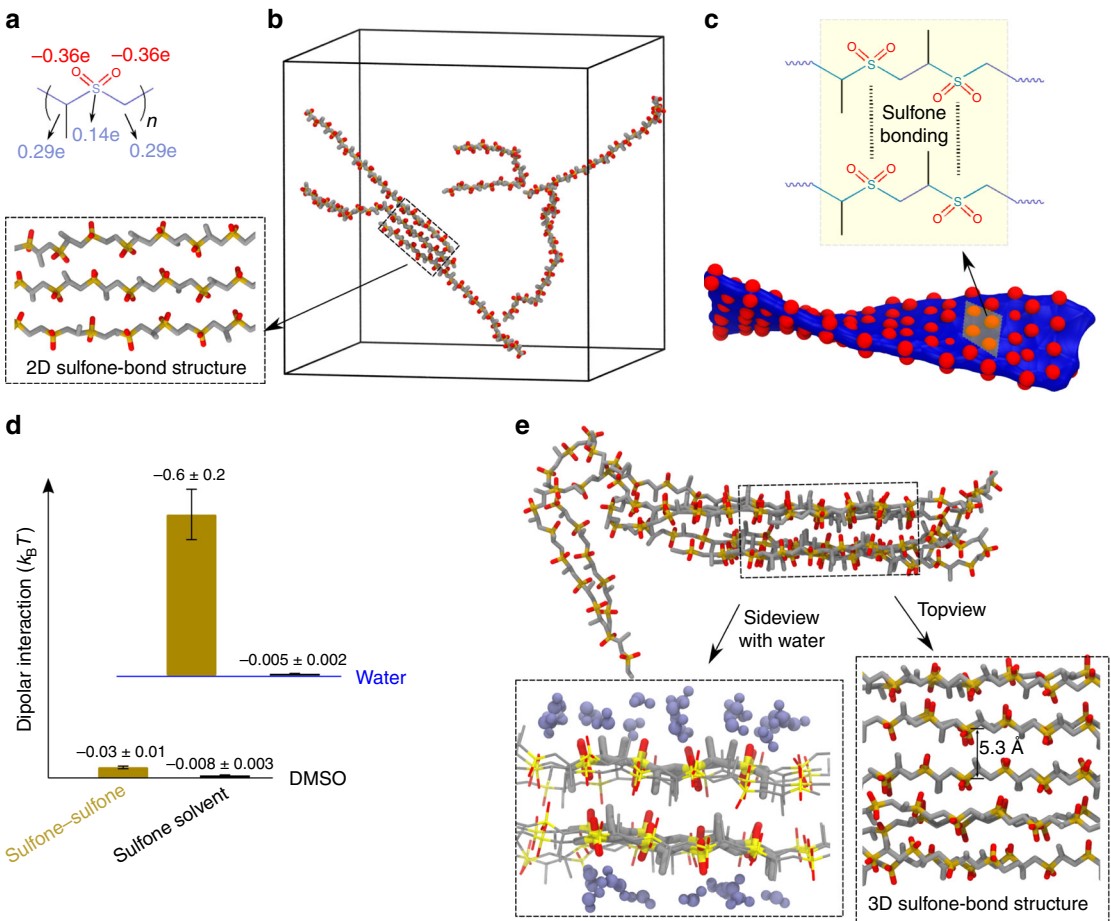

**Fig. 1 Structural features of PPSU. a** Chemical structure of PPSU showing the polymer backbone and oxygen atoms that carry positive/negative (blue/ red) atomic partial charges, respectively. **b** Atomistic simulation snapshot showing a dissolution-complementarity equilibrium in DMSO for six PPSU$_{20}$ chains. Inset is a superstructure formed by PPSU self- complementarity. **c** PPSU self-complementarity leading to a 2D reversible superstructure with enrichment of oxygen atoms on the surface. Formation of 3D superstructures is inhibited in DMSO due to the strong repulsion among layers. **d** Average dipolar energies per dipole-dipole pair of sulfone–sulfone and sulfone-solvent. Error bars represent the standard deviation from three parallel simulations. **e** Atomistic simulation snapshot showing the formation of a 3D superstructure through PPSU bundling in water. Inset showing the 3D superstructure with or without water molecules.

bonding. It is worthwhile to note that a single sulfone bond is not strong enough to create stable aggregates in aqueous solution, as indicated by the good water-solubility observed for both dimethylsulfone and random copolymers of poly(propylene sulfoxide)-*co*-poly(propylene sulfone). AAMD simulations of oligo(propylene sulfone) in water further confirm that aggregates are formed when the degree of polymerization is greater than six (Supplementary Fig. 7). Thus, we concluded that PPSU promotes the formation of sulfone-bond networks due to intra- and/or inter-chain associations.

**Large scale spatial redistribution of PPSU networks**. Because the AAMD simulations predicted that PPSU$_{20}$ can form crystalline superstructures in water (Supplementary Fig. 8), we considered whether the crystallization involved dynamic rearrangement of the sulfone–sulfone bonded network. From a kinetic point of view, network formation is preferential to crystallization given the nearly equal opportunity of all sulfones on PPSU$_{20}$ chains to form sulfone–sulfone bonds. Furthermore, the sulfone–sulfone complementarity in DMSO simulations implies that PPSU$_{20}$ tends to form sulfone–sulfone bonded networks. The gelling tendency of PPSU was experimentally confirmed by exposing DMSO solutions of PPSU$_{20}$ to humidity, which led to a sol-to-gel phase transition or precipitation respectively for high (e.g., 200 mg mL$^{-1}$) and low (e.g., 25 mg mL$^{-1}$) concentration aged PPSU$_{20}$ solutions. The resulting colorless gel or fluffy precipitates were analyzed by wide-angle X-ray diffraction (WAXD) and found to be mostly amorphous (Supplementary Fig. 9). Given the possibility that the gels in DMSO-water systems were generally under equilibrium of electrostatic repulsions and attractions, we exploited the reorganization capability for the colorless gel under the application of further hydration. By thoroughly mixing the gel with an excess amount of water, we obtained a cloudy solution in which non-uniform hydrogels of ribbon, cylindrical and spherical morphologies were found by cryogenic transmission electron microscopy (Cryo-TEM) (Supplementary Fig. 10). This drastic shape transformation from a macroscopic bulk gel to nanostructured hydrogels requires large scale spatial redistribution of PPSU chains, which involves breaking and reforming of sulfone–sulfone bonds into tightly cross-linked structures. In the process of dialysis to remove residual DMSO, we observed a further shape transformation of the nanoscale hydrogels into smaller spherical, vesicular and cylindrical morphologies (Supplementary Fig. 10). This system mimics the fibrous hydrogels created by peptide-DNA conjugates and peptides[25], providing us with a 'top-down' approach for the fabrication of various nanostructured hydrogels through network rearrangement.

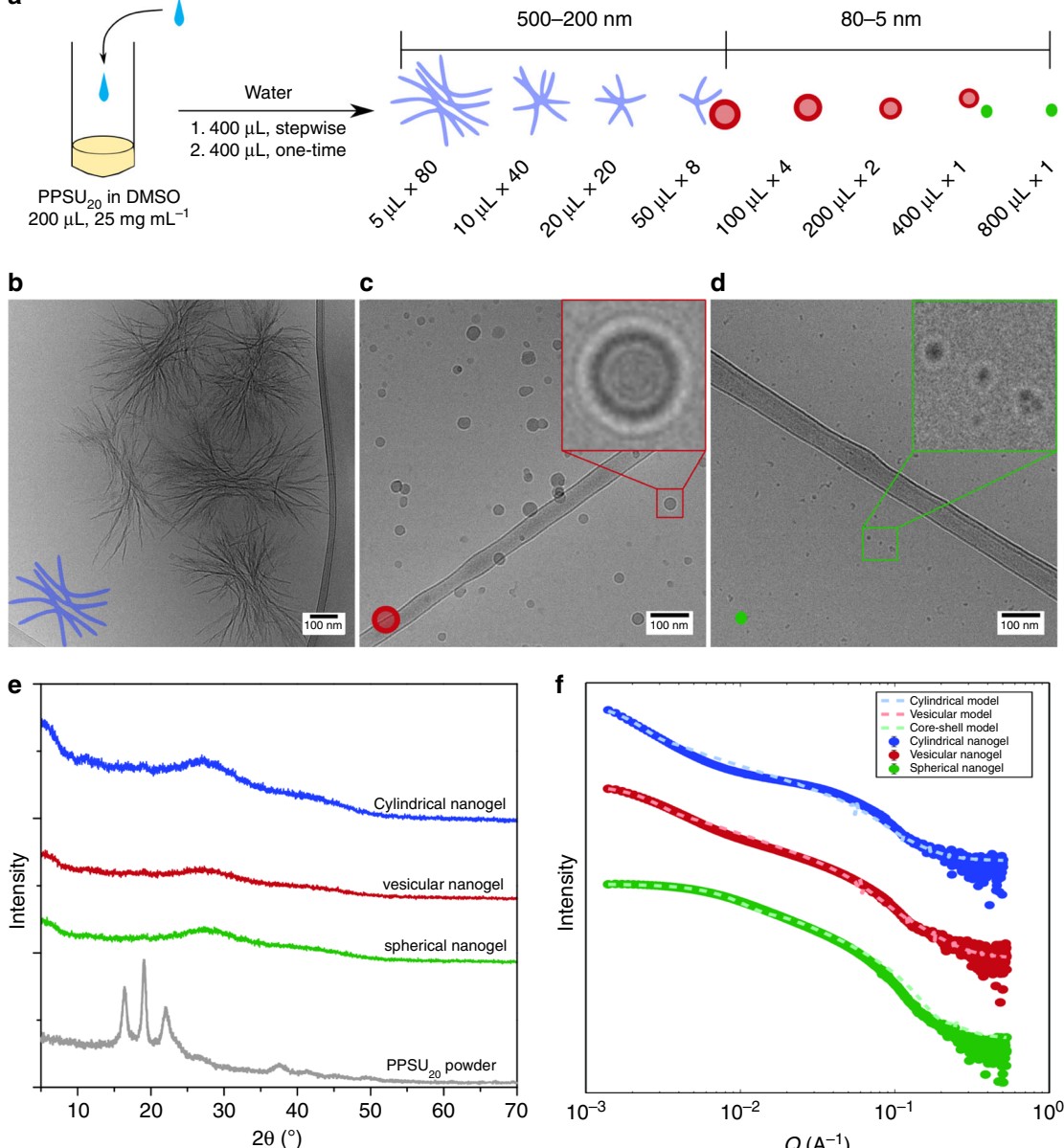

**Fig. 2 Impact of hydration history on PPSU dynamic network self-assembly. a** Fabrication of nanoscale hydrogels with distinct morphologies by homopolymer self-assembly. Externally controlling the self-assembly of PPSU$_{20}$ by stepwise addition of water to DMSO solutions of PPSU$_{20}$. **b–d** Cryo-TEM images showing the typical cylindrical, vesicular, and spherical morphologies in water. Scale bar = 100 nm. Nanostructures were prepared by stepwise hydration of PPSU$_{20}$ solutions (25 mg mL$^{-1}$ in 200 μL DMSO): **b** 5 μL per step, 80 steps; **c** 100 μL per step, 4 steps; **d** 800 μL per step, 1 step. **e** WXRD patterns of cylindrical (blue), vesicular (red) and spherical (green) nanogels, and a PPSU$_{20}$ powder comparison (gray). **f** SAXS curves of cylindrical, vesicular and spherical nanogels and corresponding model fits (cylinder, vesicle and core-shell sphere models, SasView software).

**Impact of hydration history on PPSU dynamic network self-assembly.** In an effort to control the formation of specific nanostructures, we developed a simple strategy (Fig. 2a) to fabricate all the three nanogel morphologies from the same initial DMSO solution by employing distinct hydration histories (Supplementary Fig. 11). In the strategy, we mixed DMSO solutions of PPSU$_{20}$ (25 mg mL$^{-1}$) with the same total volume of water with varying numbers of water addition steps. Cryo-TEM images revealed that larger nanobundles (Fig. 2b) formed in the case of multiple water additions while smaller spherical aggregates, including vesicular (Fig. 2c) and spherical (Fig. 2d) morphologies, were assembled using fewer hydration steps. These results were further confirmed by small-angle X-ray scattering (SAXS, Supplementary Fig. 12),

negative-stained transmission electron microscopy (Supplementary Fig. 13), energy-dispersive X-ray spectroscopy (EDS, Supplementary Fig. 13) and dynamic light scattering (DLS), with the obtained physicochemical characteristics summarized in Supplementary Table 4. No distinct patterns of PPSU crystals were observed by WXRD analysis (Fig. 2e and Supplementary Fig. 14) or SAXS (Fig. 2f), indicating amorphous structures of these nanoscale hydrogels. It is worthwhile to note that the surfaces of these nanoscale hydrogels possess enriched negative charges (Supplementary Table 4), which supports their dispersion in aqueous solution. We ascribed this phenomenon to the structural orientation of PPSU polymers, which predominantly orient their negative charges towards the surfaces of the 2D and 3D superstructures.

Furthermore, spontaneous association in aqueous solution tends to embed hydrophobic propylene spacers that bear partially positive charge, which facilitates the exposure of negative charges on surfaces. The stepwise hydration of PPSU$_{20}$ demonstrated a facile dynamic method for programmable construction of synthetic nanostructures differing in morphologies and sizes from a single-component homopolymer.

Regardless of the concentrations investigated, one-step hydration of PPSU$_{20}$ solutions in DMSO using an excess amount of water generated ultra-small (<15 nm) nanogels (Supplementary Fig. 15). These results indicate that thousands of polymer chains in an actual solution would likely entangle into interpenetrating networks following one-step hydration, effectively freezing the morphology and preventing further conformational transitions. To fabricate crystalline frameworks in water that were predicted by the AAMD simulations (Supplementary Fig. 5), cohesion among sulfones needs to be strong enough to create a stable network but not too strong to prevent dynamic exchange. In an alternative strategy, PPSU$_{20}$ in highly concentrated DMSO solutions were found to crosslink into crystalline frameworks via water diffusion. Careful addition of an excess amount of water on the top of a humidity-induced gel (Fig. 3a) resulted in the formation of a stiff crystalline solid possessing a similar WAXD pattern as that of bulk PPSU$_{20}$ powder (Fig. 3b). This result is consistent with our postulate that network formation occurs before and preferential to crystallization. The crystallization of PPSU$_{20}$ starts from a sulfone-bonded network, followed by continuous small-scale spatial redistribution of these sulfone network bonds to enable the formation of a crystalline framework

structure. This network rearrangement differs from common processes of crystallization-driven self-assembly, which involves the formation and subsequent growth of seeds by recruitment of crystalline blocks in a selective solvent[26,27].

**Mechanism of PPSU dynamic network self-assembly.** The work described above using kinetically controlled self-assembly of PPSU$_{20}$ hydrogels results in crystalline frameworks or nanostructured hydrogels of spherical, vesicular and cylindrical morphologies in aqueous solution. These hierarchical superstructures are achieved via multiple levels of sulfone organization. The first level involves the inter-chain interactions that lead to initial network formation, which is followed by a second level of sulfone–sulfone bonding that locks the polymer chains into a stable conformation. In the latter, the hydration history from multiple small additions of water leads to the bundling of PPSU$_{20}$ chains into highly organized low curvature ribbons that increase persistence length, similar to the bundling of DNA origami filaments[28]. Alternatively, quick hydration generates disordered spherical hydrogels which contain coiled PPSU$_{20}$ chains that experience greatly decreased electrostatic repulsion and strong interactions. Further hydration triggers the collapse and rearrangement of these sulfone–sulfone bonds into compact structures but imparts little change in overall chain stiffness. The formation of vesicular nanogels may be explained by taking into account the diffusivity of water into nanoscale aggregates that are stabilized by dynamic sulfone–sulfone bonding. The proximity of the outer layers of large spherical hydrogels to the water interface will promote stable crosslinking and prevent further structural rearrangement, resulting in

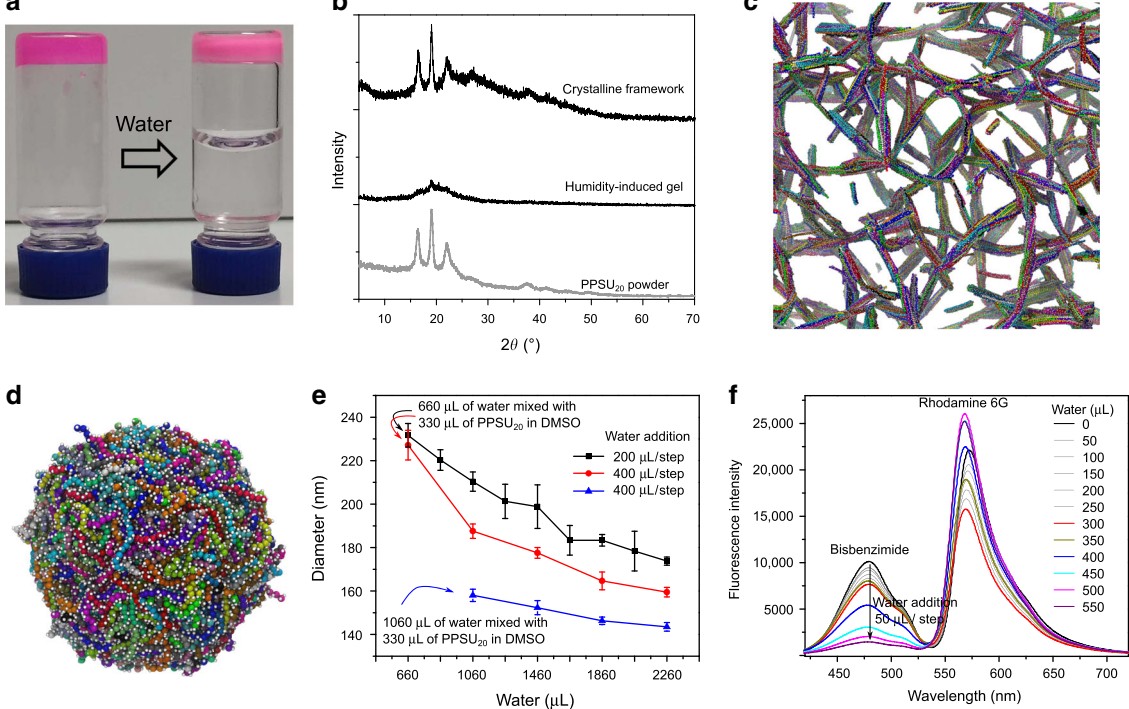

**Fig. 3 Mechanism of PPSU dynamic network self-assembly. a** Slow hydration leading to the crystalline framework. The PPSU$_{20}$ hydrogel (colored by rhodamine B) was prepared by aging a DMSO solution of PPSU$_{20}$ (200 mg mL$^{-1}$) in humidity for 3 days. Adding an excess amount of water on the top of the gel allows slow solvent replacement for dense sulfone–sulfone bonding, which induces crystallization. **b** WAXD patterns for the humidity-induce gel, crystalline framework, and for comparison the powdered solid PPSU$_{20}$. **c–d** CGMD simulation snapshots showing the formation of bundles by rigid chains and a spherical hydrogel by flexible chains. Oxygen atoms are represented by the white spheres. **e** Collapse of PPSU$_{20}$ networks as indicated by the decreasing nanogel size upon stepwise hydration. Error bars represent the standard deviation from three parallel experiments. **f** Collapse of PPSU$_{20}$ assemblies enabling FRET from bisbenzimide to rhodamine 6 G. Bisbenzimide (0.01 mg mL$^{-1}$) and rhodamine 6 G (0.01 mg mL$^{-1}$) were added to 1 mL of DMSO solution of PPSU$_{20}$ (5 mg mL$^{-1}$), followed by fluorescence titrations of the system with water (50 μL per step, $E_x = 375$ nm).

an outer shell encapsulating an inner lumen of lower density sulfone-bonding. Taken together, the diverse sulfone–sulfone bonded networks formed by $PPSU_{20}$ network self-assembly provided a strong basis to infer distinct chain conformations specified by hydration histories. We, therefore, explored through coarse-grained molecular dynamics (CGMD) simulations (Supplementary Fig. 16) the effect of chain stiffness on self-assembly. Snapshots are depicted in Fig. 3c, d, showing the formation of bundles in a framework of rigid polymer chains and a spherical, highly cross-linked hydrogel of flexible $PPSU_{20}$. To further assess dynamic changes in nanogel size upon continued hydration, we tracked $PPSU_{20}$ assemblies using DLS and observed significant decreases in diameter as the ratios of water increased in water-DMSO mixed systems for different hydration histories (Fig. 3e and Supplementary Fig. 17). Fluorescence titrations of $PPSU_{20}$ assemblies with water were also performed to assess the collapse of these polymer networks using the Förster resonance energy transfer (FRET) pair of bisbenzimide and rhodamine 6G (Supplementary Fig. 18). In Fig. 3f, we first observed the fluorescence intensity of both dyes to decrease upon water dilution, followed by a FRET signal from bisbenzimide to rhodamine 6G. The FRET efficiency improved upon further hydration, suggesting decreasing sizes and thus collapse of $PPSU_{20}$ aggregates. These results demonstrate a compressing process for $PPSU_{20}$ networks that achieves smaller structures as water ratios increase.

**Molecular capture capability of PPSU nanogels.** The sulfone–sulfone bonded networks of $PPSU_{20}$ are mimetic of electrostatic networks in dimethylsulfone, a high-temperature polar solvent that can dissolve a wide range of organic solutes and is miscible with many other solvents. We investigated the use of $PPSU_{20}$ nanostructured hydrogels to capture organic molecules from DMSO or aqueous solution using the stepwise hydration strategy (Supplementary Movie 5). The results are shown in Table 1, wherein exceptionally high encapsulation efficiencies (EE) of >95% were observed for a wide range of molecules, including hydrophobic Nile red and water-soluble fluorescein isothiocyanate (FITC), doxorubicin hydrochloride (DOX/HCl), dextran, green fluorescent protein (GFP), DNA, and RNA. We also investigated the loading of FITC-labeled albumin (FITC-BSA), a model protein, at varying concentrations (Supplementary Fig. 19). The rapid water-induced collapse of $PPSU_{20}$ networks allowed encapsulation of nearly 100% of protein molecules from aqueous solutions with up to 80% encapsulation capacity (w/w) (Table 1, entry 5 to 7). Over 80% EE was achieved when nanoparticles contained 20% more protein than polymer by mass (Table 1, entry 8).

$PPSU_{20}$ nanostructures were highly stable in water and no aggregation or precipitation were observed when aging a suspension of vesicular nanogels for 20 days at room temperature (Supplementary Table 5, Supplementary Fig. 20). When containing payloads, no premature payload release was observed in water after days of incubation at room temperature (Supplementary Fig. 21). Given the possibility of breaking the nanostructures using less polar solvents, we proceeded to investigate the stability of $PPSU_{20}$ vesicular nanogels in tetrahydrofuran (THF)-water mixed systems. Rhodamine B (Rh B) was encapsulated in the nanogels to evaluate the degradation. We followed the leakage of Rh B in these systems and observed that the nanogels are less stable in THF-water mixed solvents than in water or in THF (Supplementary Fig. 21). This result is consistent with our conclusion that sulfone–sulfone interactions are susceptible to solvent polarity.

**Mechanism of molecular capture by PPSU nanogels.** The high stability of PPSU nanogels in water provides insight into the mechanism of drug encapsulation. Starting from a swollen network that shows high affinity for a wide range of organic solutes,

**Table 1 Encapsulation efficiency of molecules captured during $PPSU_{20}$ self-assembly[a].**

| Entry | Molecule[b] | Mass ratio (drug/$PPSU_{20}$) | Encapsulation efficiency (%)[c] | |
|---|---|---|---|---|
| | | | 100 μL per step | 10 μL per step |
| 1 | Nile red | 0.004 | >98 | >96 |
| 2 | FITC | 0.004 | >99 | >99 |
| 3 | DOX/HCl | 0.08 | >97 | >96 |
| 4 | Dextran | 0.08 | >96 | >96 |
| 5 | Albumin | 0.08 | >99 | >99 |
| 6 | Albumin | 0.4 | >99 | >99 |
| 7 | Albumin | 0.8 | >96 | >96 |
| 8 | Albumin | 1.2 | 80 | 83 |
| 9 | GFP[d] | 0.016 | >99 | >99 |
| 10 | RNA | 0.016 | >96 | >98 |
| 11 | DNA | 0.016 | >96 | >97 |

[a]100 μL of aqueous drug (except Nile red) solutions were mixed stepwise with 50 μL of $PPSU_{20}$ solution (25 mg mL$^{-1}$ in DMSO), followed by one-time quick hydration using 400 μL of water. Nile red was loaded using its DMSO solution.
[b]GFP = Recombinant *A. victoria* GFP protein. Dextran ($M_w$ = 4000), Albumin, RNA, and DNA are conjugated with FITC.
[c]Encapsulation efficiency is defined as the ratio of the number of molecules in the assemblies to the total amount applied in the formulation. The encapsulation efficiencies were calculated by fluorescence measurements.
[d]GFP has no detectable fluorescence after encapsulation.

subsequent hydration induces quick collapse of the sulfone–sulfone bonded network (Fig. 3c–f), in which the diffusivity of encapsulated molecules decreases significantly once they are incorporated into the nanogel. We inferred that such a molecular trapping mechanism should lead to physicochemical changes to the nanogels upon encapsulation due to the properties of the payload. To investigate the trapping of protein within PPSU nanogels without interference from nonspecific protein adsorption, we employed FITC-BSA, which possesses a net negative charge that would minimize interactions with the negatively charged PPSU nanogel surfaces (Supplementary Fig. 22). When $PPSU_{20}$ assembly was induced by a concentration series of aqueous FITC-BSA solution instead of pure water (Fig. 4a), we found that both nanogel diameter and zeta potential were influenced by FITC-BSA (Supplementary Table 6). TEM images on negatively stained samples (Fig. 4b–e and Supplementary Fig. 23) revealed that while a low ratio of FITC-BSA (0.1 mg mL$^{-1}$, 1.6% w/w protein/PPSU) to PPSU had no impact on morphology (Fig. 4b, c), increasing the concentration of FITC-BSA to 1.0 mg mL$^{-1}$ (16% w/w protein/PPSU) induced formation of macroscale aggregates consisting of vesicular nanogel complexes (Fig. 4d). Under the application of an even greater FITC-BSA concentration (10 mg mL$^{-1}$, 160% w/w protein/PPSU), we observed increased aqueous dispersibility and decreased size and polydispersity for the resulting nanogels with no evidence of macroscale gel formation (Fig. 4e, Supplementary Table 6). Of note, loading of FITC-BSA at 0.1 mg mL$^{-1}$ and 1.0 mg mL$^{-1}$ achieved ~100% EE, and an impressive 45% EE was achieved in the presence of 10 mg mL$^{-1}$ of protein, which generated nanogels possessing 72% protein (3.6 mg FITC-BSA/5 mg PPSU) by mass (Fig. 4f).

These results suggest that FITC-BSA is accessible on the surface of the nanogels at densities dependent upon the initial loading concentration, and one potential explanation for this is described in the schematic of Fig. 4a. The lower concentration (0.1 mg mL$^{-1}$) resulted in minimal surface exposure of FITC-BSA, while the intermediate (1.0 mg mL$^{-1}$) loading concentration resulted in a partial surface exposure, which may have allowed sharing of protein between different nanogels during PPSU network collapse to induce aggregation (Fig. 4d). Since free FITC-BSA molecules have limited

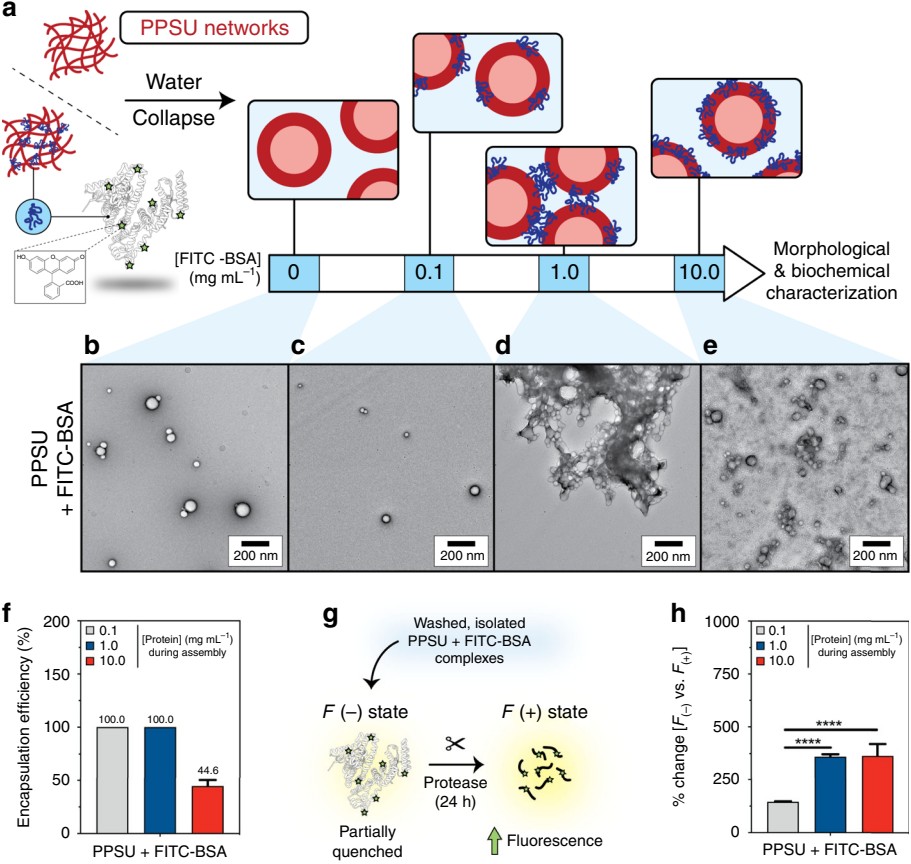

**Fig. 4 Encapsulation of protein in PPSU nanogels. a** PPSU$_{20}$ nanostructures (5 mg) were assembled with a concentration series of aqueous FITC-BSA at 0.1, 1.0, or 10 mg mL$^{-1}$ (1.6, 16, and 160% w/w protein/PPSU). PPSU$_{20}$ nanostructures prepared without protein (0 mg mL$^{-1}$ FITC-BSA) were included as a control. **b**–**e** Transmission electron microscopy of negatively stained specimens. **b** PPSU$_{20}$ nanostructures absent of protein. PPSU-FITC-BSA complexes assembled in the presence of (**c**) 0.1, (**d**) 1.0, or (**e**) 10.0 mg mL$^{-1}$ FITC-BSA (purposefully overloaded sample and the free FITC-BSA was not removed). Scale bar = 200 nm. **f** Encapsulation efficiency of FITC-BSA protein cargo. **g** Trypsin proteolysis assay illustration. In the absence of protease, FITC-BSA is partially quenched and exhibits weak fluorescence. After trypsin treatment, FITC-labeled peptides are released and the fluorescence of the FITC fluorophore increases. **h** The percentage increase in FITC fluorescence (i.e., 100 × ($F_{(+)}$ − $F_{(−)}$)/$F_{(−)}$) after trypsin proteolysis was calculated, where $F_{(+)}$ is the fluorescence after protease treatment and $F_{(−)}$ refers to the fluorescence in the absence of protease treatment. Increased FITC fluorescence is a readout of whether embedded protein interfaces with the external aqueous environment. Error bars represent the standard deviation from three parallel experiments. Statistical significance was determined by Tukey's multiple comparisons test. ****$p < 0.0001$.

influence on nanogels aggregation (Supplementary Fig. 24), the saturated loading concentration (10.0 mg mL$^{-1}$) likely resulted in a high surface density of exposed FITC-BSA, which prevented aggregation between nanogels (Fig. 4e). Due to its net negative charge, BSA is not extensively denatured on negatively charged surfaces and has been predicted to orient with negative residues and domains exposed to the aqueous solution after adsorption[29,30]. This is supported by our zeta potential analysis shown in Supplementary Table 6 and indicates that adsorbed BSA would not induce the PPSU nanogel aggregation observed in Fig. 4d. We employed trypsin digestion to assess payload accessibility, and ~30–35% of encapsulated FITC-BSA was accessible to cleavage for the overloaded 10 mg mL$^{-1}$ sample (Fig. 4g, h and Supplementary Fig. 25), suggesting considerable solution exposure of loaded protein.

We concluded that the observed high molecular encapsulation achieved by PPSU is due to tight molecular trapping, which allows us to fabricate protein-based nanocarriers without requiring covalent chemistries. By adjusting formulation conditions, nanogels up to 120% protein by mass could be achieved (Table 1). Our results thus demonstrate the possibility of extensive customization for the formation of protein/PPSU nanogels. The decreased fluorescence of GFP upon loading suggests that the current

methodology may partially denature the protein structure, likely due to exposure to DMSO during loading (Table 1). Further investigations will be required to determine the limits of this platform for protein loading and retention of bioactivity as well as to assess the specific influences of different protein physicochemical properties on the final products.

## Discussion

We demonstrate that sulfone–sulfone bonded networks have exceptional molecular capture capability, providing a drug delivery platform that would greatly enhance the ability to encapsulate therapeutic molecules. Although there have been intensive studies on delivery via self-assembled nanocarriers in the past few decades, substantial challenges remain due in part to their low loading of water-soluble biologics and hydrophilic small molecule therapeutics. Drug loading in polymeric vesicles (i.e., polymersomes) for example are typically lower than 20%; some are even substantially less than 5%[31]. For their potential use in biomedical applications, we further demonstrated no cytotoxicity up to 0.25 mg mL$^{-1}$ (Supplementary Figs. 26 and 27) and high cellular uptake (Supplementary Figs. 28 and 29) of PPSU$_{20}$ nanogels in macrophages,

with intracellular delivery being mediated by both macropinocytosis and clathrin-mediated endocytosis mechanisms (Supplementary Fig. 30). Macrophages were selected for the study due to their high phagocytic activity, allowing us to investigate both receptor-mediated and receptor-independent mechanisms of uptake. Furthermore, macrophages are the primary cells in the body responsible for the clearance of nanoparticles due to their high prevalence in the liver and spleen.

In summary, we demonstrate that PPSU promotes the formation of an electrostatic network via sulfone–sulfone bonds in DMSO-water mixed systems. The sulfone–sulfone bonded network undergoes kinetically dependent self-assembly upon hydration, which imparts control over the morphology and size of macro- and nanoscale hydrogels. Without the need to change polymer structure or molecular weight, this single-component homopolymer system permits fabrication of crystalline frameworks and nanoscale hydrogels of spherical, vesicular, and cylindrical morphologies, controlled easily by the addition of water. The PPSU network shows a high affinity for a wide range of organic molecules, similar to the high-temperature solvent dimethyl sulfone. Taking advantage of this capability, we are able to encapsulate diverse hydrophilic molecules with nearly 100% efficiency, including large biologics like protein and DNA that are often difficult to load. These results demonstrate that noncovalent sulfone–sulfone inter-and intra-chain bonding between PPSU homopolymers presents a structurally dynamic synthetic system that is biomimetic of natural biopolymers. PPSU is therefore a versatile platform for supramolecular chemistry that may have extensive applications in areas requiring efficient molecular encapsulation, such as biomedicine.

## Methods

**Polymer synthesis.** $PPS_{20}$ was prepared by anionic ring-opening polymerization of propylene sulfide (20 equiv.) using ethanethiol (1 equiv.) initiator and benzyl bromide (5 equiv.) end-capper in the presence of sodium methylate as a base in dry dimethyl formamide[32]. Mixing $PPS_{20}$ with 30% of hydrogen peroxide (1 g $PPS_{20}$ per 100 mL of $H_2O_2$ solution) and shaking the mixtures led to a homogeneous solution overnight. Lyophilization of the obtained solution resulted in shiny solids of $PPSU_{20}$ without the requirement for further purification.

**AAMD simulations.** Classical all-atom MD simulations were performed using the CHARMM 36 m force field[33]. The recommended CHARMM TIP3P water model[34] was applied with the structures constrained using the SETTLE algorithm[35]. The simulations were performed using the package GROMACS (version 2016.3)[36]. A detailed description of the simulation procedure is provided in the Supplementary Materials. The short-range electrostatic interactions were calculated up to 1.2 nm, and the long-range electrostatic interactions were calculated by means of the Particle Mesh Ewald algorithm[37]. A time step of 2 fs (2.5 fs) was employed by constraining all the covalent bonds using the LINCS algorithm[38] in the DMSO system (water system). Annealing simulations were performed to speed up the convergence of the equilibrations[39].

**Dipolar energy calculation.** By following a previous work[40], we calculated the dipolar interactions between the neighbor units. Each sulfone is defined as one charge-neutral unit, as well as one DMSO molecule and one $H_2O$ molecule. The calculation of dipolar interaction energy between the charge-neutral units is provided in the Supplementary Materials. The average dipole moments were calculated to be 2.347 $D$ for the CHARMM TIP3P water model, the same as the reported value of 2.347 $D$[41]. The dipole moment of DMSO was calculated to be 5.22 $D$, in consistent with the reported value of 5.11 $D$ in the original literature where the DMSO CHARMM force field were originally presented, and around 20% larger than the experimental value[42]. The dipole moment was calculated to be 6.534 $D$ for the sulfone.

**Fabrication of $PPSU_{20}$ hydrogels.** Two hundred microliters of $PPSU_{20}$ solutions (25 mg mL$^{-1}$ in DMSO) were added stepwise with 400 μL of water and then one-time with another 400 μL of water. Each step was followed by vortexing to thoroughly mix the samples. After dialysis, the nanogels were applied for CryoTEM imaging, SAXS, DLS, WAXD, and tetrazolium (MTT) assay.

**CGMD simulations.** Each polymer is modeled by a linear bead-spring chain consisting of $N = 20$ coarse-grained (CG) monomers. Each CG monomer carries three-point charges: a positive charge in the backbone corresponding to the S atoms and two neighboring backbone C atoms, and two negative charges corresponding to the O atoms (white spheres). Overall the monomer is charge neutral and has a net dipole as indicated by the all-atom model calculations. The relative positions of the three-point charges are maintained by harmonic springs between the S–O bonds and constraining the O–S–O angle. The non-bonded interactions between the CG monomers include the excluded volume interaction and electrostatics. The excluded volume interaction is modeled by the Weeks–Chandler–Andersen potential, which is the Lennard–Jones potential truncated and shifted to zero at the minimum. The parameter is chosen as the length unit of the system, which corresponds to the van der Waals diameter of the S atom. The electrostatic interaction is truncated at the cutoff distance of $r_c = 8\sigma$, which is considered adequate for our CG simulations where the solvent is treated implicitly as a uniform back ground with the dielectric constant of 47 for DMSO and 80 for water. The parameters of the CG model are calibrated against the all-atom in terms of the chain persistent length in these two solvents. To ensure that such a truncated Coulombic scheme does not affect our results, we have also performed test simulations with long-range Coulombic electrostatics with the particle–particle particle–mesh method. The simulations with long-range Coulombic electrostatics yield similar assembled morphologies. All the CGMD simulations were performed with the LAMMPS (version 19 September 2019) software package[43].

**Loading experiments.** For the loading of Nile red, nanogels were prepared using pure water and Nile red-containing (0.1 mg mL$^{-1}$) DMSO solutions of $PPSU_{20}$. For the loading of water-soluble drugs, aqueous solutions of FITC (0.05 mg mL$^{-1}$), DOX/HCl (1.0 mg mL$^{-1}$), dextran (1.0 mg mL$^{-1}$), albumin (various concentrations), GFP (0.2 mg mL$^{-1}$), RNA (0.2 mg mL$^{-1}$), DNA (0.2 mg mL$^{-1}$), and DMSO solutions of $PPSU_{20}$ (25 mg mL$^{-1}$) were used. Typically, 100 μL of the corresponding aqueous solution were mixed stepwise (10 μL per step or 100 μL per step) with 50 μL of $PPSU_{20}$ solution under vortex, then another 400 μL of water was added. After centrifugation at 16,000 × g for 10 min, the fluorescence of the supernatant was measured and the EE were calculated.

**Trypsin digestion of FITC-BSA-loaded $PPSU_{20}$ nanogels.** DMSO solutions of $PPSU_{20}$ (200 μL, 25 mg mL$^{-1}$) were hydrated using 800 μL of aqueous FITC-BSA solutions (0, 0.1, 1.0, 10.0 mg mL$^{-1}$). DMSO was removed by dialysis after self-assembly. Negatively stained FITC-BSA-loaded $PPSU_{20}$ vesicular nanogels were characterized by TEM. After removal of unloaded FITC-BSA by three rounds of centrifugation and resuspension in phosphate-buffered saline, FITC-BSA-loaded $PPSU_{20}$ vesicular nanogels were subjected to digestion with trypsin protease (2.0 μg mL$^{-1}$; Trypsin Gold, Promega) for 24 h at 37 °C, 80 rpm. Undigested FITC-BSA-loaded $PPSU_{20}$ vesicular nanogels (i.e., no protease treatment) were included as a control. After 24 h incubation with trypsin, FITC fluorescence ($E_x = 490$ nm, $E_m = 525$ nm) was quantified in triplicate using a SpectraMax M3 microplate reader (Molecular Devices). The percentage increase in FITC fluorescence (i.e., $100 \times (F_{(+)} - F_{(-)})/F_{(-)}$) after trypsin proteolysis was calculated, where $F_{(+)}$ is the fluorescence after protease treatment and $F_{(-)}$ is referred to the fluorescence in the absence of protease treatment.

**Reporting summary.** Further information on research design is available in the Nature Research Reporting Summary linked to this article.

## Data availability
All relevant data are available from authors upon request.

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

# ARTICLE

8. Görl, D., Zhang, X., Stepanenko, V. & Würthner, F. Supramolecular block copolymers by kinetically controlled co-self-assembly of planar and core-twisted perylene bisimides. *Nat. Commun.* **6**, 7009 (2015).
9. Chen, S. & Binder, W. H. Dynamic ordering and phase segregation in hydrogen-bonded polymers. *Acc. Chem. Res.* **49**, 1409–1420 (2016).
10. Boyken, S. E. et al. De novo design of protein homo-oligomers with modular hydrogen-bond network–mediated specificity. *Science* **352**, 680 (2016).
11. Wojtecki, R. J., Meador, M. A. & Rowan, S. J. Using the dynamic bond to access macroscopically responsive structurally dynamic polymers. *Nat. Mater.* **10**, 14–27 (2011).
12. Pappas, C. G. et al. Dynamic peptide libraries for the discovery of supramolecular nanomaterials. *Nat. Nanotechnol.* **11**, 960–967 (2016).
13. Zhang, F., Nangreave, J., Liu, Y. & Yan, H. Structural DNA nanotechnology: state of the art and future perspective. *J. Am. Chem. Soc.* **136**, 11198–11211 (2014).
14. Napoli, A., Valentini, M., Tirelli, N., Muller, M. & Hubbell, J. A. Oxidation-responsive polymeric vesicles. *Nat. Mater.* **3**, 183–189 (2004).
15. Napoli, A., Tirelli, N., Kilcher, G. & Hubbell, A. New synthetic methodologies for amphiphilic multiblock copolymers of ethylene glycol and propylene sulfide. *Macromolecules* **34**, 8913–8917 (2001).
16. Karabin, N. B. et al. Sustained micellar delivery via inducible transitions in nanostructure morphology. *Nat. Commun.* **9**, 624 (2018).
17. Du, F., Liu, Y.-G. & Scott, E. A. Immunotheranostic polymersomes modularly assembled from tetrablock and diblock copolymers with oxidation-responsive fluorescence. *Cell. Mol. Bioeng.* **10**, 357–370 (2017).
18. Du, F., Bobbala, S., Yi, S. & Scott, E. A. Sequential intracellular release of water-soluble cargos from shell-crosslinked polymersomes. *J. Controlled Release* **282**, 90–100 (2018).
19. Vasdekis, A. E., Scott, E. A., O'Neil, C. P., Psaltis, D. & Hubbell, J. A. Precision intracellular delivery based on optofluidic polymersome rupture. *ACS Nano* **6**, 7850–7857 (2012).
20. Yu, W., He, X., Vanommeslaeghe, K. & MacKerell, A. D. Jr. Extension of the CHARMM general force field to sulfonyl-containing compounds and its utility in biomolecular simulations. *J. Comput. Chem.* **33**, 2451–2468 (2012).
21. Bordwell, F. G. & Cooper, G. D. The effect of the sulfonyl group on the nucleophilic displacement of halogen in α-halo sulfones and related substances1. *J. Am. Chem. Soc.* **73**, 5184–5186 (1951).
22. Clark, T., Murray, J. S., Lane, P. & Politzer, P. Why are dimethyl sulfoxide and dimethyl sulfone such good solvents? *J. Mol. Modeling* **14**, 689–697 (2008).
23. Lowe, A. B. & McCormick, C. L. Synthesis and solution properties of Zwitterionic polymers. *Chem. Rev.* **102**, 4177–4190 (2002).
24. Teng, P. et al. Hydrogen-bonding-driven 3D supramolecular assembly of peptidomimetic zipper. *J. Am. Chem. Soc.* **140**, 5661–5665 (2018).
25. Freeman, R. et al. Reversible self-assembly of superstructured networks. *Science* **362**, 808 (2018).
26. Gilroy, J. B. et al. Monodisperse cylindrical micelles by crystallization-driven living self-assembly. *Nat. Chem.* **2**, 566 (2010).
27. Choi, I., Yang, S. & Choi, T.-L. Preparing semiconducting nanoribbons with tunable length and width via crystallization-driven self-assembly of a simple conjugated homopolymer. *J. Am. Chem. Soc.* **140**, 17218–17225 (2018).
28. Castro, C. E., Su, H.-J., Marras, A. E., Zhou, L. & Johnson, J. Mechanical design of DNA nanostructures. *Nanoscale* **7**, 5913–5921 (2015).
29. Kubiak-Ossowska, K., Jachimska, B. & Mulheran, P. A. How negatively charged proteins adsorb to negatively charged surfaces: a molecular dynamics study of BSA adsorption on silica. *J. Phys. Chem. B* **120**, 10463–10468 (2016).
30. Kubiak-Ossowska, K., Tokarczyk, K., Jachimska, B. & Mulheran, P. A. Bovine serum albumin adsorption at a silica surface explored by simulation and experiment. *J. Phys. Chem. B* **121**, 3975–3986 (2017).
31. Allen, S., Osorio, O., Liu, Y.-G. & Scott, E. Facile assembly and loading of theranostic polymersomes via multi-impingement flash nanoprecipitation. *J. Controlled Release* **262**, 91–103 (2017).
32. Cerritelli, S., Velluto, D. & Hubbell, J. A. PEG-SS-PPS: reduction-sensitive disulfide block copolymer vesicles for intracellular drug delivery. *Biomacromolecules* **8**, 1966–1972 (2007).
33. Huang, J. et al. CHARMM36m: an improved force field for folded and intrinsically disordered proteins. *Nat. methods* **14**, 71 (2016).
34. MacKerell, A. D. et al. All-atom empirical potential for molecular modeling and dynamics studies of proteins. *J. Phys. Chem. B* **102**, 3586–3616 (1998).
35. Miyamoto, S. & Kollman, P. A. Settle: an analytical version of the SHAKE and RATTLE algorithm for rigid water models. *J. Comput. Chem.* **13**, 952–962 (1992).
36. Hess, B., Kutzner, C., van der Spoel, D. & Lindahl, E. GROMACS 4: algorithms for highly efficient, load-balanced, and scalable molecular simulation. *J. Chem. Theory Comput.* **4**, 435–447 (2008).
37. Essmann, U. et al. A smooth particle mesh Ewald method. *J. Chem. Phys.* **103**, 8577–8593 (1995).
38. Hess, B. P-LINCS: a parallel linear constraint solver for molecular simulation. *J. Chem. Theory Comput.* **4**, 116–122 (2008).
39. Ortony, J. H. et al. Water dynamics from the surface to the interior of a supramolecular nanostructure. *J. Am. Chem. Soc.* **139**, 8915–8921 (2017).
40. Qiao, B., Ferru, G., Olvera de la Cruz, M. & Ellis, R. J. Molecular origins of mesoscale ordering in a metalloamphiphile phase. *ACS Cent. Sci.* **1**, 493–503 (2015).
41. Mark, P. & Nilsson, L. Structure and dynamics of the TIP3P, SPC, and SPC/E water models at 298 K. *J. Phys. Chem. A* **105**, 9954–9960 (2001).
42. Strader, M. L. & Feller, S. E. A flexible all-atom model of dimethyl sulfoxide for molecular dynamics simulations. *J. Phys. Chem. A* **106**, 1074–1080 (2002).
43. Plimpton, S. Fast parallel algorithms for short-range molecular dynamics. *J. Comput. Phys.* **117**, 1–19 (1995).

## Acknowledgements
The authors are grateful to Jonathan Remis for cryo-TEM observation. We thank the support from the Center for Computation & Theory of Soft Materials, the BioCryo facility of Northwestern University's NUANCE Center, the Integrated Molecular Structure Education and Research Center, Structural Biology Facility, NU Atomic, the Nanoscale Characterization Experimental Center, Robert H. Lurie Comprehensive Cancer Center Flow Cytometry Core, and Biological Imaging Facility at Northwestern University. SAXS experiments were performed at the DuPont-Northwestern-Dow Collaborative Access Team (DND-CAT) located at Sector 5 of the Advanced Photon Source (APS). DND-CAT is supported by Northwestern University, E.I. DuPont de Nemours & Co., and The Dow Chemical Company. This research used resources of the Advanced Photon Source, a U.S. Department of Energy (DOE) Office of Science User Facility operated for the DOE Office of Science by Argonne National Laboratory under Contract No. DE-AC02-06CH11357. This work was supported by the National Institutes of Health Director's New Innovator Award (grant no. 1DP2HL132390-01), the National Institute of Allergy and Infectious Diseases (grant no. 5R21AI137932-02), the National Science Foundation CAREER Award (grant no. 1453576), the Louis A. Simpson & Kimberly K. Querrey Center for Regenerative Nano-medicine Regenerative Nanomedicine Catalyst Award, and the Department of Energy Award DE-FG02-08ER46539, and the Sherman Fairchild Foundation.

## Author contributions
F.D. designed and contributed to all the experiments, analyzed data and wrote the manuscript; B.Q. designed performed the AAMD simulation and wrote the manuscript; T.N. performed the CGMD simulations; M.V. contributed to the analysis of material interactions with protein, the TEM characterization of negatively stained specimens, cell experiments, and the preparation of figures and illustrations; S.B. contributed to the SAXS measurements and the cell experiments; S.Y. contributed to the cell experiments; C.L. contributed to EDS; V.D., M.O.d.l.C., and E.S. supervised the research and wrote the manuscript.

## Competing interests
The authors declare no competing interests.
