## [Peer Review File · Nature Communications]

Reviewers' comments:

Reviewer #1 (Remarks to the Author):

In this work, the authors use an experimental, theoretical, and computer simulation framework to study the process of kinetically controlled assembly of polypropylene sulfone (PPSU) into crystals or nanostructured hydrogels.

The authors found interesting behavior of polymer network collapse based on hydration history that could lead to the development of materials for water purification as well as encapsulation and delivery of biologics. I think the results are interesting but I would like to see the following concerns/questions addressed:

1. In the abstract, the authors state that achieving the hierarchic assembly behavior seen in natural biomolecules using synthetic polymers remains elusive. I would like to suggest that this statement is deflated. There is plenty of work using synthetic polypeptides for instance that can form protein-like α -helices. In addition, there are a number of reports on rather complex biomimetic structures made of abiological polymeric systems.
2. It is not true that only amphiphilic block-copolymers show hierarchic self-assembly. What is true is that normally you need two blocks that are "immiscible" in order for hierarchic structures to form. The fact that in this case such behavior is seen in a PPSU homopolymer through dynamic sulfone-sulfone bonding is indeed intriguing.
3. Related to point two, the authors discuss the effect of electrostatics on the assembly behavior so it would be worth discussing what is the degree of intrinsic amphiphilicity of PPSU as well as the possibility of "charge density polydispersity". I bring this up because this system collapses upon addition of water. That behavior is typical in associative polymer phase behavior.
4. Figure 2a: why do the tubular nano gels adopt a spherulite morphology? This is normally an indication of polymer crystallinity.
5. Figure 2d: the intent of this plot is to show that the nano gel structures do not show crystallinity. That is fine but I think that the figure would be much more informative and less confusing if it included the SAXS form factor fitting (now at SI) of the nano gel shapes AND the corresponding XRD "amorphous data". Putting a reference XRD profile of a crystalline PPSU system would be good.
6. I am a little bit confused (so maybe this should be clarified with a cartoon) of what the nano gel sample is like. Are the vesicles, cylinders, etc in a gel-state and these particles are then dispersed in water? If one removes the water after the formation of the nanogels, would the system crystallize or would one have a bulk gel with some mesophase supramolecular structure? Maybe this experiment should be done. This would help infer about the thermodynamic stability of the system and reversibility.
7. The cell uptake experiments should be clarified. Aren't macrophages up-taking "anything"? What is the mechanism of cell entry?

Reviewer #2 (Remarks to the Author):

In this work, Scott and coworkers report on a homopolymer that undergoes network assembly and collapse in aqueous dimethylsulfoxide solutions. Their work aimed to mimic the self-assembly found in Nature and, specifically, sought to control self-assembly of single component

supramolecular systems to address a long-standing challenge for synthetic polymers. The work is fundamentally interesting, the simulations are well supported by experimental results, the scholarly presentation and broad interest are high. The authors report on the self-assembly and inter/intramolecular interactions and their influence on nanostructure formation using simulations and experiment; however, these results are not particularly surprising. Other work has reported similar phenomenon but with different conditions and structure formations (Scientific Reports, 2017, 7, Article number: 13138; Macromol. Chem. Phys. 2013, 214, 143–158; Nature Materials, 2004, 3, 183–189). Overall, the science and conclusions are adequately supported by the data and the literature references are sufficient; however, it is the opinion of this reviewer that the overall impact of this work is only moderate to high and has the potential to be readily appreciated by the readership of Nature Communications with a few experimental additions and revisions to elevate the overall impact of the work. Below I list some comments, concerns, and suggestions regarding this work:

1. The authors report testing a wide variety of solvents for the PPSU20 crystals, yet only find success with DMSO for dissolving the PPSU20 crystals. Did the authors work with 1-Methyl-2-pyrrolidinone (NMP), which has similar solvation properties to DMSO? This is an interesting industrial solvent that is easier to make and keep anhydrous and has not been widely investigated in the context of this polymer. The use of NMP and its effects on the polymer and its self-assembly properties may be worth exploring.

2. To further assess dynamic changes in nanogel size upon continued hydration, the authors tracked PPSU20 assemblies using DLS and fluorescent titrations. ¹H DOSY NMR spectroscopy would corroborate their results and bolster their conclusions. Furthermore, the authors report that they are able to encapsulate diverse hydrophilic molecules, including often difficult to load large biologics such as proteins and DNA; however, the authors don't elaborate on the long-term stability of such systems. Do they leak cargo over time and at what rate? Are the entrapped cargo stable over time? Stability measurements over longer timeframes would provide interesting information on the long-term stability of this system. It would also be interesting to further probe the effect of charge and functional groups when investigating molecular cargo for encapsulation. Currently, there is little discussion on the mechanism of capture / uptake of the molecular cargo. Can the authors elucidate?

3. The authors rely heavily on their simulations to support their conclusions. It would help the reader considerably if more detail elaborating on the significance, considerations or implications of their simulations were included in the manuscript. The authors should provide more detail on the conditions and significance of their cluster formation analysis (Figure S4 provides little detail on the model). Additionally, it would be interesting to know distances between chains (in Fig. 1) within the sulfone-based polymer. Also, where are the solvent molecules located in the molecules? How do they solvate the PPSU20 crystals? It would be nice to see an image in Fig 1 depicting the solvation.

Minor comments:

- Figure 1 caption has extra space between (c) and (d)
- The authors should change all "ul" and "µl" unit designations to "µL" (see SI, page 7, in the section entitled "Transmission electron microscopy of negatively stained PPSU20 structures")
- Table S1 in the SI should include the full name of DMSO

Reviewer #3 (Remarks to the Author):

Here the authors present a unique homopolymer (PPSU) that self-assemble upon water titration or polymer DMSO solutions. Interestingly, specific PPSU structures were formed upon specific water dilution protocols; the structures were thoroughly analyzed with molecular simulations. The

authors also demonstrated that small molecules and proteins can be encapsulated in the PPSU structures. The work is interesting and novel but requires further context and experimental details. Therefore, I recommend major revisions prior to publications, as listed below.

The study does lack context for the average reader with respect to practical limitations. For example, for drug encapsulation a direct discussion about the mechanism of drug encapsulation may demonstrate the improvement over other self-organizing polymers and amphiphilic polymers. The MS would benefit greatly from better context, the authors should clearly state how this MS is a clear advancement, either through application or through fundamental polymer design and assembly.

Prior to publication, the following comments should be addressed:

- The stability of the nanostructures in different buffers or solvents should be discussed or investigated. Would physiological salt concentrations disrupt or change the nanostructures?
- Please add error bars to Fig. 1d.
- Drug encapsulation. Please discuss the mechanism of drug encapsulation, especially with respect to hydrophilic vs hydrophobic molecules. Why do the authors believe encapsulation was so high? For protein encapsulation, the authors should differentiate between encapsulated proteins and proteins non-specifically adsorbed to the surface of nanostructures. The authors should also encapsulate a bioactive protein and determine if the encapsulation process reduces bioactivity. Furthermore, the authors should investigate if encapsulated drugs are released from the nanostructures over time.
- For intracellular uptake, please provide a rationale for the use of RAW264.7 macrophages.
- The authors state there is limited toxicity from using an MTT assay, however some samples only resulted in 80% viability, which may indicate some cell death and toxicity. A live/dead stain should be performed to conclude if cell death is occurring.
- Fig. S10 may be added to the main text to better explain the protocol for nanostructure formation.
- On page 16, what does "m/m" stand for? Mass/mass?
- Are there any limitations on polymer molecular weight?
- What is the dispersity of the nanostructures form? Can it be quantified?

Reviewers' comments:

Reviewer #1 (Remarks to the Author):

In this work, the authors use an experimental, theoretical, and computer simulation framework to study the process of kinetically controlled assembly of polypropylene sulfone (PPSU) into crystals or nanostructured hydrogels.

The authors found interesting behavior of polymer network collapse based on hydration history that could lead to the development of materials for water purification as well as encapsulation and delivery of biologics. I think the results are interesting but I would like to see the following concerns/questions addressed:

1. In the abstract, the authors state that achieving the hierarchic assembly behavior seen in natural biomolecules using synthetic polymers remains elusive. I would like to suggest that this statement is deflated. There is plenty of work using synthetic polypeptides for instance that can form protein-like alfa-helices. In addition, there are a number of reports on rather complex biomimetic structures made of abiological polymeric systems.

*The reviewer makes an excellent point, and we have therefore rewritten the abstract as well as parts of the introduction accordingly. Instead of “elusive”, we now state that synthetic self-assembling homopolymer systems are “limited”. We also specify that we are referring to bio/nanomaterial applications, as we do acknowledge the wide range of synthetic systems amenable to hierarchic assembly that are not specifically applicable to biological/medical applications.

Abstract: “Natural biomolecules such as peptides and DNA can dynamically organize into diverse hierarchical structures. Mimicry of this homopolymer self-assembly using synthetic systems has remained limited but would be advantageous for the design of adaptive bio/nanomaterials.”

Introduction: “ Such controlled aqueous self-assembly of single component supramolecular systems has remained a challenge for synthetic polymers, which typically require amphiphilicity or direct incorporation of naturally occurring peptide/nucleic acid monomers or derivatives thereof for controlled aggregation in aqueous systems.”

2. It is not true that only amphiphilic block-copolymers show hierarchic self-assembly. What is true is that normally you need two blocks that are “immiscible” in order for hierarchic structures to form. The fact that in this case such behavior is seen in a PPSU homopolymer through dynamic sulfone-sulfone bonding is indeed intriguing.

*We thank the reviewer for pointing out this detail and we are very glad to hear that they find our sulfone-sulfone bonding mechanism to be intriguing. The reviewer is correct, so we have updated the text in the introduction for clarity. We now present the case of amphiphilic copolymers as just one common example, avoiding any misinterpretation that we are stating that *only* amphiphilic block copolymers show hierarchic assembly.

“The most common synthetic self-assembling systems employ amphiphilic copolymers that are limited to separate hydrophobic and hydrophilic segments, which cannot allow the complex dynamic self-assembly behavior seen in nature while also maintaining aggregate stability required for practical applications.”

3. Related to point two, the authors discuss the effect of electrostatics on the assembly behavior so it would be worth discussing what is the degree of intrinsic amphiphilicity of PPSU as well as the possibility of “charge density polydispersity”. I bring this up because this system collapses upon addition of water. That behavior is typical in associative polymer phase behavior.

*We thank the reviewer for the suggestion. As a homopolymer containing only a single monomer unit, PPSU is not amphiphilic. Accordingly, we experimentally found that short PPSU chains ($DP < 5$) are water soluble, and the simulations also suggested that even shorter PPSU chains (for example DP of 2 and 4) are water soluble (Fig. S7). These results imply intrinsic hydrophilicity of PPSU, due to the polar structure. High molecular weight PPSU results in increased chain self-associations via sulfone-sulfone bonding (which induces the phase transition), similar to some zwitterionic polymers (Ref. 23).

The following text was added to clarify:

“AAMD simulations of oligo(propylene sulfone) in water further confirm that aggregates formed when the degree of polymerization is bigger than 6 (Fig. S7).”

(Ref. 23) A. B. Lowe, C. L. McCormick, Synthesis and Solution Properties of Zwitterionic Polymers. *Chemical Reviews* 102, 4177 (2002).

4. Figure 2a: why do the tubular nano gels adopt a spherulite morphology? This is normally an indication of polymer crystallinity.

We appreciate the reviewer for raising this point. Our results indicate the absence of crystallinity for the nanobundles (Fig 2e); therefore we do not believe that these are spherulites. Slow hydration via exposure to humidity facilitates crystallization, as demonstrated in Fig. 3; however, the tubular nanogels were fabricated using a comparatively rapid multiple-step water addition method that limited crystallinity. We inferred that the nanobundles were formed by collapse of ribbons shown in Fig. S10. During this transition, while some parts of the nanogels were still connected, those completely divided parts would repel each other due to the negative charges (Fig. 1c). This could be the reason that the nanobundles adopt a spherulite morphology. It is clear that this interesting phenomenon will require further study, so we would rather not speculate as to the mechanism behind the formation of this spherulite morphology in this current manuscript. We kindly request the opportunity to investigate this mechanism in a more focused future study.

5. Figure 2d: the intent of this plot is to show that the nano gel structures do not show crystallinity. That is fine but I think that the figure would be much more informative and less confusing if it included the SAXS form factor fitting (now at SI) of the nano gel shapes AND the corresponding XRD “amorphous data”. Putting a reference XRD profile of a crystalline PPSU system would be good.

*Thank you very much for these excellent suggestions. We have included the SAXS results as well as a reference XRD profile of PPSU crystal in Figure 2.

6. I am a little bit confused (so maybe this should be clarified with a cartoon) of what the nano gel sample is like. Are the vesicles, cylinders, etc in a gel-state and these particles are then dispersed in water? If one removes the water after the formation of the nanogels, would the system crystallize or would one have a bulk gel with some mesophase supramolecular structure? Maybe this experiment should be done. This would help infer about the thermodynamic stability of the system and reversibility.

*We apologize for this confusion and have included a schematic in Fig.2 that was previously in the supplement, which shows the PPSU nanogel formation methodology as well as the diverse morphologies that result under different conditions. We have also added a scheme in Fig. 4 to show the collapse from networks to vesicles. The nanogels are well-dispersed in aqueous solution as evidenced by our TEM images and DLS data (Fig 2, 3e, 4, S11, &Table S5).

We did try to remove water after the formation of the nanogels, but this was not possible without disrupting the assembled PPSU superstructure. After attempting direct removal of water by lyophilization, we obtained amorphous solids which contain residual water and DMSO. After further heating the amorphous solids under vacuum, the white sample became black before complete removal of the residual solvents, suggesting significant chemical modification. Complete removal of water from PPSU nanogels was therefore achieved by dissolving the nanogels in DMSO (which disrupted all PPSU superstructure), followed by lyophilization. Of note, for samples treated this way, we can reform nanogels by hydration of the resulting DMSO solution of PPSU, further verifying that the self-assembly and gelation are due to noncovalent interactions.

7. The cell uptake experiments should be clarified. Aren't macrophages up-taking "anything"? What is the mechanism of cell entry?

*We agree with the reviewer that it would be instructive to investigate the mechanism of cell entry. Cell uptake experiments were thus performed to determine whether macropinocytosis and clathrin-mediated endocytosis mechanisms contribute to the internalization of PPSU₂₀ vesicular nanogels and nanobundles. These are two key mechanisms of cellular entry, the former being receptor independent and the latter being receptor mediated. RAW 264.7 macrophages were pre-treated with pharmacological inhibitors of macropinocytosis and clathrin-mediated endocytosis prior to administering PPSU nanostructures. Specifically, macropinocytosis was inhibited by cellular pre-treatment with cytochalasin D (5 μ M) for 2 h, whereas clathrin-dependent endocytosis was inhibited using chlorpromazine (100 μ M) for 30 min. Cells pre-treated with PBS were included as a control. After this pre-treatment period, cells were incubated with 0.25 mg/mL FITC-dextran loaded PPSU₂₀ vesicular nanogels or nanobundles, and flow cytometry was performed to quantify uptake. Both cytochalasin D and chlorpromazine pre-treatment significantly decreased uptake of PPSU₂₀ nanovesicles and bundles (Fig. S27). This result suggests that both macropinocytosis and clathrin-mediated endocytosis contribute to the internalization of PPSU nanostructures. Of note, we intended for this first publication on PPSU to focus on the assembly mechanism and as the first demonstration of molecular encapsulation.

A full analysis of biological applications and controlled delivery is beyond the scope of this current work, so we have added the methods for these cell experiments to just the supplement.

The following was added to the main text to summarize these findings:

“For their potential use in biomedical applications, we further demonstrated low cytotoxicity (Figs. S24 and S25) and high cellular uptake (Figs. S26 to S27) of PPSU20 nanogels in macrophages, with intracellular delivery being mediated by both macropinocytosis and clathrin-mediated endocytosis (Fig. S28).”

Reviewer #2 (Remarks to the Author):

In this work, Scott and coworkers report on a homopolymer that undergoes network assembly and collapse in aqueous dimethylsulfoxide solutions. Their work aimed to mimic the self-assembly found in Nature and, specifically, sought to control self-assembly of single component supramolecular systems to address a long-standing challenge for synthetic polymers. The work is fundamentally interesting, the simulations are well supported by experimental results, the scholarly presentation and broad interest are high. The authors report on the self-assembly and inter/intramolecular interactions and their influence on nanostructure formation using simulations and experiment; however, these results are not particularly surprising. Other work has reported similar phenomenon but with different conditions and structure formations (Scientific Reports, 2017, 7, Article number: 13138; Macromol. Chem. Phys. 2013, 214, 143–158; Nature Materials, 2004, 3, 183–189). Overall, the science and conclusions are adequately supported by the data and the literature references are sufficient; however, it is the opinion of this reviewer that the overall impact of this work is only moderate to high and has the potential to be readily appreciated by the readership of Nature Communications with a few experimental additions and revisions to elevate the overall impact of the work. Below I list some comments, concerns, and suggestions regarding this work:

We thank the reviewer for these supportive comments. In response to the provided list of similar phenomena, we believe that our system is entirely unique compared to this prior work. Since all of the reviewer’s referenced papers describes systems employing poly(propylene sulfide) (PPS) we feel the need to kindly clarify that we are not using PPS in this study. PPS is a commonly employed hydrophobic oxidation sensitive block for many self-assembling systems (many of which are used by the Prof. Evan Scott lab). Here, we are instead using poly(propylene **sulfone**) (PPSU), which is the fully oxidized form of PPS. In this work, we describe how self-assembly of PPSU (hydrophilic, not oxidation sensitive, not incorporated into a copolymer) employs sulfone-sulfone bonding, which is a completely different mechanism than that observed for self-assembling systems employing PPS (water insoluble, oxidation responsive, requires a hydrophilic block for self-assembly via PPS aggregation). Of note, PPSU homopolymer self-assembly has never before been published, and previously it was well hypothesized [Ref. 14] that PPSU was completely water-soluble, and not supportive of controlled aggregation.

(Ref. 14) A. Napoli, M. Valentini, N. Tirelli, M. Muller, J. A. Hubbell, Oxidation-responsive polymeric vesicles. Nature materials 3, 183 (2004).

Below is a brief summary of why our PPSU system is quite different than these prior publications:

Scientific Reports, 2017, 7, Article number: 13138: Hydrogel fabrication in this publication required a dual system composed of natural and synthetic components to form ROS-responsive microparticles. This system employed PPS mixed with collagen, a natural polymer. Firstly, our system did not use PPS. We employed PPSU, which is the oxidized and extremely hydrophilic derivative of PPS. Furthermore, a critical advantage and difference of our system is that we only required PPSU homopolymer, and no other components, to self-assemble diverse nanostructures. This publication required collagen and an emulsion polymerization methodology. Also, PPSU is not ROS-responsive, as it is already fully oxidized, and we focused our manuscript on the self-assembly of nanoparticles, not the emulsion polymerization of microparticles as described in this work.

Macromol. Chem. Phys. 2013, 214, 143–158: This is a review article that covers oxidation-responsive polymers, with extensive discussion of PPS-based systems. The properties of PPSU (not oxidation responsive) and the mechanism of PPSU homopolymer self-assembly are not related to the work discussed in this review.

Nature Materials, 2004, 3, 183–189: This is one of the first papers discussing the self-assembly and ROS-sensitivity of PEG-*b*-PPS self-assembling systems. This work describes a block copolymer system using PPS as the hydrophobic block. The main focus of our current manuscript is on *homopolymer* self-assembly of PPSU. A key distinction and advantage of our work is that a block copolymer is not required, only the PPSU homopolymer is necessary for self-assembly. Also, this prior *Nature Materials* publication does not discuss PPSU self-assembly. PPSU is only mentioned as a potential product of PPS that is generated after PPS is oxidized. Oxidation of PPS in the *Nature Materials* publication actually generates mixed polymers containing both sulfone and sulfoxide groups, (further verified in Vasdekis et al. *ACS Nano* 2013), so PPSU is actually not being generated in that publication.

1. The authors report testing a wide variety of solvents for the PPSU₂₀ crystals, yet only find success with DMSO for dissolving the PPSU₂₀ crystals. Did the authors work with 1-Methyl-2-pyrrolidinone (NMP), which has similar solvation properties to DMSO? This is an interesting industrial solvent that is easier to make and keep anhydrous and has not been widely investigated in the context of this polymer. The use of NMP and its effects on the polymer and its self-assembly properties may be worth exploring.

We thank the reviewer for raising this interesting point. We tested the solubility of PPSU₂₀ in NMP and it is correct that NMP can also break PPSU₂₀ crystals into a clear solution. NMP is not as effective as DMSO for dissolving PPSU₂₀: the solubility of PPSU₂₀ in 1 mL of NMP is ~23 mg, while 1 mL of DMSO can dissolve >250 mg PPSU₂₀. We have included this result in Table S1. Similar hydration history-dependent self-assembly behavior was observed upon hydration of NMP solutions of PPSU₂₀. When we mixed a DMSO or NMP solution of PPSU₂₀ with the same total volume of water employing a simple strategy that involved adjusting only the number of water addition steps, well-dispersed nanogels were obtained where the transmittance decreased as the number of water addition steps increased. Further investigation and optimization of PPSU self-assembly to achieve uniform nanogels using an NMP system will require a large number of experiments and cryo-TEM characterization, which

extend beyond the scope of our present work using a DMSO system. However, we acknowledge that NMP may also serve as an alternative to DMSO when high concentrations of PPSU are not required, and we therefore greatly appreciate the reviewer's suggestion. We have added this new NMP data to Table S1.

2. To further assess dynamic changes in nanogel size upon continued hydration, the authors tracked PPSU₂₀ assemblies using DLS and fluorescent titrations. ¹H DOSY NMR spectroscopy would corroborate their results and bolster their conclusions. Furthermore, the authors report that they are able to encapsulate diverse hydrophilic molecules, including often difficult to load large biologics such as proteins and DNA; however, the authors don't elaborate on the long-term stability of such systems. Do they leak cargo over time and at what rate? Are the entrapped cargo stable over time? Stability measurements over longer timeframes would provide interesting information on the long-term stability of this system. It would also be interesting to further probe the effect of charge and functional groups when investigating molecular cargo for encapsulation. Currently, there is little discussion on the mechanism of capture / uptake of the molecular cargo. Can the authors elucidate?

Thank you very much for the suggestion of using ¹H DOSY NMR spectroscopy to assess dynamic changes in nanogel size upon continued hydration. Due to the COVID-19 pandemic, currently, the NMR spectrometer in Integrated Molecular Structure Education and Research Center (IMSERC) at Northwestern University is not accessible for us to perform the training and experiments. We believe that DOSY experiments would be an excellent option for future experiments to investigate the assembly of PPSU, but is beyond the scope of our current manuscript.

We acknowledge the importance of the cargo capture mechanism and stability assessments and have significantly modified the manuscript, most notably through the addition of a new figure. 4. PPSU₂₀ nanostructures were found to be highly stable under physiological conditions, however, our preliminary cell release experiments revealed successful intracellular delivery of various cargoes such as doxorubicin hydrochloride and vaccine antigens. This combination of stability and intracellular bioresponsive release suggests that PPSU may be a powerful platform for controlled delivery applications. Currently, we are trying to figure out the intracellular breaking mechanism of PPSU nanogels, and we would like to share some data that is also consistent with the conclusion that sulfone-sulfone interactions are susceptible to solvent polarity. We have investigated the stability of PPSU₂₀ vesicular nanogels in tetrahydrofuran (THF)-water mixed systems to exploit the possibility of breaking the nanostructures using less polar solvents. In these experiments, rhodamine B (Rh B) was encapsulated in the nanogels to evaluate the degradation. We followed the leakage of Rh B in these systems and observed that the nanogels are less stable in THF-water mixed solvents than in water or in THF (Fig. S20). Upon 3-day incubation in water, no premature payload release of Rh B was observed in water (Fig. S20), indicating irreversible encapsulation which allows high encapsulation of payloads by PPSU nanogels.

To investigate the encapsulation mechanism, we employed FITC-BSA, a negatively charged protein, which we selected to prevent nonspecific interactions with the negatively charged PPSU surfaces. First, we incubated PPSU₂₀ nanogels with FITC-labeled albumin (FITC-BSA), and the results revealed that FITC-BSA did not adsorb after the formation of nanostructures. We then considered whether the

encapsulation involves molecular trapping which allows stable embedding of payloads in collapsed PPSU networks. This encapsulation mechanism was confirmed by the self-assembly of PPSU₂₀ with a FITC-BSA concentration series (Fig. 4).

Furthermore, we have utilized three model proteins with distinct isoelectric points: albumin, hemoglobin and lysozyme to assess the influence of charge on both protein adsorption and loading and the results revealed > 99% encapsulation efficiency for all three proteins. Because positively charged proteins can non-specifically adsorb to the negatively charged surface of nanostructures, we only focused on the encapsulation of BSA in this manuscript to avoid confusion. Furthermore, we intended for this manuscript to focus on the basic process of self-assembly of PPSU, and additional loading data dilutes this message. Of note, we are currently working on a follow-up manuscript that will investigate the loading of these and many other proteins, biomolecules and small molecules in detail. Due to the focus of this current manuscript on the fundamental assembly of PPSU via polarity driven sulfone-sulfone interactions, we kindly request the opportunity to have a future drug encapsulation and delivery publication that covers this application of PPSU in sufficient detail.

3. The authors rely heavily on their simulations to support their conclusions. It would help the reader considerably if more detail elaborating on the significance, considerations or implications of their simulations were included in the manuscript. The authors should provide more detail on the conditions and significance of their cluster formation analysis (Figure S4 provides little detail on the model). Additionally, it would be interesting to know distances between chains (in Fig. 1) within the sulfone-based polymer. Also, where are the solvent molecules located in the molecules? How do they solvate the PPSU₂₀ crystals? It would be nice to see an image in Fig 1 depicting the solvation.

We thank the reviewer for raising these questions and providing us the opportunity to clarify. The cluster formation analysis is an established method to quantify the aggregation behavior of molecules (Ref. S1). When the maximum cluster size of 1 is reached, no aggregation occurs; otherwise, the molecules are forming aggregates with a favored size. In our calculations, we calculated the inter-polymer cluster formation. Specifically, when any sulfur atom on PPSU chain A is within 6.7Å from any sulfur on chain B, these two PPSU chains A and B are defined as one cluster. Here 6.7Å is the upper distance of the first inter-polymer S-S radial distribution functions (Fig. S8). As demonstrated in Fig. S4, the maximum probability occurs for the cluster size of 1 in all three parallel simulations, it thus supports that the PPSU chains are not favoring aggregation. That is, they are soluble in the DMSO solvent.

We extended the caption of Fig. S4 to include more details:

Distribution of PPSU₂₀ clusters in the three parallel all-atom simulations in DMSO. PPSU₂₀ chains are viewed as clusters if the distance of any inter-PPSU chain sulfur atoms is less than 0.67 nm (the first minimum in the radial distribution function in Fig. S8). A maximum probability occurs at the cluster size of 1 supports that the PPSU chains are dispersed in DMSO solvent (Ref. S1). The GROMACS program *gmx clustersize* was employed for the calculations.

Regarding the distance between PPSU chains (Fig. 1), the optimal distance is found to be 5.3 Å based on the primary peak of the inter-PPSU chain S-S radial distribution function (Fig. S8). This distance is

now provided in Fig. 1. Note that the upper distance of the first inter-PPSU chain S-S interactions is 6.7 Å, which was employed for the cluster formation analysis as discussed above.

As suggested by the reviewer, we also included a new plot on the PPSU hydration in Fig. 1e. It is shown that the neighboring water molecules are located above and below the PPSU aggregates, but not inside the PPSU aggregate. Therefore, the PPSU chains are highly dehydrated inside the aggregates.

The following text was added for clarity:

“Neighboring water molecules were predicted to be located above and below but not inside of the PPSU aggregates (Fig. 1e).”

(Ref. S1) B Qiao, KC Littrell, RJ Ellis, Phys. Chem. Chem. Phys. 20, 12908-12915

Minor comments:

- Figure 1 caption has extra space between (c) and (d)
- The authors should change all “ul” and “μl” unit designations to “μL” (see SI, page 7, in the section entitled “Transmission electron microscopy of negatively stained PPSU20 structures”)
- Table S1 in the SI should include the full name of DMSO

We thank the reviewer for finding these issues and have corrected them accordingly.

Reviewer #3 (Remarks to the Author):

Here the authors present a unique homopolymer (PPSU) that self-assemble upon water titration or polymer DMSO solutions. Interestingly, specific PPSU structures were formed upon specific water dilution protocols; the structures were thoroughly analyzed with molecular simulations. The authors also demonstrated that small molecules and proteins can be encapsulated in the PPSU structures. The work is interesting and novel but requires further context and experimental details. Therefore, I recommend major revisions prior to publications, as listed below.

The study does lack context for the average reader with respect to practical limitations. For example, for drug encapsulation a direct discussion about the mechanism of drug encapsulation may demonstrate the improvement over other self-organizing polymers and amphiphilic polymers. The MS would benefit greatly from better context, the authors should clearly state how this MS is a clear advancement, either through application or through fundamental polymer design and assembly.

Thank you very much for the suggestion. We investigated the encapsulation mechanism and concluded that payloads are trapped in the nanogels (see the response to comment 2 of reviewer 2, Fig. 4). From a view of biomedical applications, the molecular trapping mechanism allows the loading of payloads with higher efficiency and versatility than the vast majority of alternative strategies. To the best of our knowledge, we are not aware of any alternative nanoparticle system capable of loading the range of payloads listed in Table 1 all with >95% efficiency or with such a simple protocol. We have added this

statement in the main text to show the advancement of this system:

“We concluded that the observed high molecular encapsulation achieved by PPSU is due to tight molecular trapping, which allows us to fabricate novel protein-based nanocarriers without requiring covalent chemistries. By adjusting formulation conditions, nanogels up to 120% protein by mass could be achieved (Table 1). Our results thus demonstrate the possibility of extensive customization for the formation of protein/PPSU nanogels.”

“Our results demonstrate that sulfone-sulfone bonded networks have immense potential for applications in water purification, nanomedicine and diagnostics. In comparison, vesicles formed from amphiphile assembly typically show low encapsulation efficiency (<20%) for biologics and hydrophilic small molecule therapeutics (29). For their potential use in biomedical applications, we further demonstrated low cytotoxicity (Figs. S24 and S25) and high cellular uptake (Figs. S26 to S27) of PPSU20 nanogels in macrophages, with intracellular delivery being mediated by both macropinocytosis and clathrin-mediated endocytosis mechanisms (Fig. S28).”

Prior to publication, the following comments should be addressed:

- The stability of the nanostructures in different buffers or solvents should be discussed or investigated. Would physiological salt concentrations disrupt or change the nanostructures?

We thank the reviewer for these suggestions. We acknowledge the importance of stability assessments and response to this comment can be found above for the response to comment 2 of reviewer 2. The nanostructures are stable in water, phosphate-buffered saline (PBS), and aqueous BSA solution (Fig. 4e, the TEM image was obtained in the presence of unloaded BSA).

- Please add error bars to Fig. 1d.

We thank the reviewer for finding this issue. We have added error bars to Fig. 1d.

- Drug encapsulation. Please discuss the mechanism of drug encapsulation, especially with respect to hydrophilic vs hydrophobic molecules. Why do the authors believe encapsulation was so high? For protein encapsulation, the authors should differentiate between encapsulated proteins and proteins non-specifically adsorbed to the surface of nanostructures. The authors should also encapsulate a bioactive protein and determine if the encapsulation process reduces bioactivity. Furthermore, the authors should investigate if encapsulated drugs are released from the nanostructures over time.

Thank you very much for the suggestion. We investigated the encapsulation mechanism and concluded that payloads are trapped in the nanogels (see our response to comment 2 of reviewer 2, Fig. 4). Such an encapsulation mechanism allows high encapsulation of both hydrophilic and hydrophobic molecules. As the first demonstration of protein encapsulation, we incubated FITC-BSA with PPSU nanogels after the formation of nanostructures and this protein was not found to non-specifically adsorb to the surface of nanostructures. In our view, this result is not surprising given the negative surface charge for both BSA and PPSU nanogels. We actually selected this protein due to its net negative charge, which allowed

us to investigate molecular encapsulation without interference from nonspecific protein adsorption. To address these issues, Figure S21 was added to the manuscript along with the following text:

“To investigate the trapping of protein within PPSU nanogels without interference from nonspecific protein adsorption, we employed FITC-BSA, which possesses a net negative charge that would minimize interactions with the negatively charged PPSU nanogel surfaces (Fig. S21).”

In terms of bioactivity following encapsulation, the decreased fluorescence of GFP upon loading suggests that the current methodology may partially denature the protein structure, likely due to exposure to DMSO during loading (Table 1). Admittedly, the quenching of GFP fluorescence could also be a result of increased concentration due to the compression of the PPSU network into compact nanogels as verified by the FRET experiments in Fig. 3f and S18). In our following publication, we will further assess the non-specifically adsorption of PPSU nanogels with various proteins and develop method to retain the activity of proteins. When containing drugs, PPSU₂₀ nanostructures were highly stable and no leakage was detected over time in water (Fig. S20). Further discussion of these findings were included in the text and presented ***as a new Fig. 4. Five new paragraphs*** were added to the end of the manuscript to describe this new data. All changes in the manuscript are marked in red.

- For intracellular uptake, please provide a rationale for the use of RAW264.7 macrophages.

The reviewer has made an excellent suggestion. We selected macrophages due to their highly phagocytic nature, allowing us to investigate both receptor mediated and receptor independent endocytosis. Furthermore, macrophages are the primary cells in the body responsible for clearance of nanoparticles due to their prevalence in the liver and spleen. We have added the following text to the manuscript to clarify this rationale:

“Macrophages were selected for the study due to their highly phagocytic nature, allowing us to investigate both receptor mediated and receptor independent endocytosis. Furthermore, macrophages are the primary cells in the body responsible for the clearance of nanoparticles due to their prevalence in the liver and spleen.”

- The authors state there is limited toxicity from using an MTT assay, however some samples only resulted in 80% viability, which may indicate some cell death and toxicity. A live/dead stain should be performed to conclude if cell death is occurring.

We thank the reviewer for the suggestion. We performed live/dead staining to investigate cell viability upon nanostructures treatment. The result is shown in Fig. S24, wherein macrophage cell viability following nanostructure treatment was determined using Zombie Aqua fixable cell viability dye. The results are consistent with the MTT assay, showing limited toxicity of PPSU nanogels.

- Fig. S10 may be added to the main text to better explain the protocol for nanostructure formation.

The reviewer makes an excellent point that showing the protocol for nanostructure formation in the main text would be better. We have updated Fig. 2 to show the protocol.

- On page 16, what does “m/m” stand for? Mass/mass?

We apologize for the confusion. We have changed m/m into w/w which is commonly used.

- Are there any limitations on polymer molecular weight?

We did explore both experimentally and through simulations the effect of PPSU molecular weight. It was found that stable aggregates can form when the degree of polymerization (DP) is bigger than 6 (shown in Fig. S7). To eliminate the influence of polymer end groups, we experimentally used higher molecular weight PPSU ($DP_{\text{NMR}} = 20, 33, 57$) for self-assembly. For all these polymers, similar self-assembly results were obtained (Note the polymers are homopolymers). We did also try to use even bigger DP, for which completely oxidation became difficult and their aggregates in water became much smaller.

The following text was added to clarify:

“AAMD simulations of oligo(propylene sulfone) in water further confirmed that aggregates formed when the degree of polymerization is bigger than 6 (Fig. S7).”

- What is the dispersity of the nanostructures form? Can it be quantified?

Based on the results of CryoTEM and negative-staining TEM, we can obtain uniform nanostructures for each of the cylindrical, vesicular, and spherical morphologies (Fig. S11). Polydispersity index (PDI) in dynamic light scattering (DLS) for vesicular, and spherical morphologies is below 0.3 (Table S4 and S5).

REVIEWER COMMENTS

Reviewer #1 (Remarks to the Author):

The authors made significant changes to the manuscript text and figures to clarify the main findings and impact of their work. This reviewer considers the paper relevant to the community and has no further concerns to raise.

-Cecilia Leal

Reviewer #2 (Remarks to the Author):

The authors have sufficiently addressed all of my comments and concerns. I recommend this work be published in Nature Communications.

Reviewer #3 (Remarks to the Author):

The authors have conducted a great deal of additional work, and improved the manuscript. To reach the impact level of nature communications, greater mechanistic understanding of potential applications should be included, as described below.

- 1) The authors states that "Our results demonstrate that sulfone-sulfone bonded networks have immense potential for applications in water purification, nanomedicine and diagnostics". This statement does not appear to be fully supported by the data. For example, it is unclear how these materials would be useful for water purification or diagnostics. It is also unclear if the materials can be used for protein delivery (nanomedicine) because the bioactivity of the proteins was not confirmed. Therefore, more results need to be included to demonstrate applications or these statements need to be removed.
- 2) Stability. Show stability of particles over time using a technique such as DLS, to demonstrate that the particles are stable as described.
- 3) Figure S24-25. The materials appear to be cytotoxic at higher concentrations. The non-cytotoxic concentration range should be better defined, and statistics should be performed.
- 4) Figure S21. In the measurement of fluorescent BSA adsorption onto particles surfaces, the authors followed the fluorescence of the supernatant. No decrease in fluorescence was observed. It would be better to follow the fluorescence of the particles directly, as the decrease in supernatant fluorescence may not be significant even with protein adsorption.
- 5) Figure 4H. Why was SEM and not standard deviation used for panel H? Is the y-axis in panel H equal to $F(+)/F(-) * 100$? The statement "with a 5% significance level" is not required and confusing.
- 6) Figure 4. The reason for particle aggregation as a function of BSA concentration is not clear. Is it due to BSA-BSA interactions? Is it due to the denaturation of BSA, which exposes sticky hydrophobic domains? If so, does very high BSA concentrations prevent aggregation due to free BSA in solution? A better explanation is required.

Reviewer #1 (Remarks to the Author):

The authors made significant changes to the manuscript text and figures to clarify the main findings and impact of their work. This reviewer considers the paper relevant to the community and has no further concerns to raise.

-Cecilia Leal

We thank the reviewer very much for the supportive comments.

Reviewer #2 (Remarks to the Author):

The authors have sufficiently addressed all of my comments and concerns. I recommend this work be published in Nature Communications.

We thank the reviewer very much for the supportive comments.

Reviewer #3 (Remarks to the Author):

The authors have conducted a great deal of additional work, and improved the manuscript. To reach the impact level of nature communications, greater mechanistic understanding of potential applications should be included, as described below.

1) The authors states that "Our results demonstrate that sulfone-sulfone bonded networks have immense potential for applications in water purification, nanomedicine and diagnostics". This statement does not appear to be fully supported by the data. For example, it is unclear how these materials would be useful for water purification or diagnostics. It is also unclear if the materials can be used for protein delivery (nanomedicine) because the bioactivity of the proteins was not confirmed. Therefore, more results need to be included to demonstrate applications or these statements need to be removed.

We appreciate the supportive comments from the reviewer. We would kindly like to note that we intended for this first manuscript to mainly focus on PPSU's novel form of homopolymer self-assembly, and thus we presented extensive computational and experimental analysis of the mechanism behind this process. There exists a wide range of applications for a platform that can encapsulate such a wide range of molecules at a high encapsulation efficiency, and it would be impossible to adequately address each of these in our first manuscript on PPSU. We have therefore selected drug encapsulation as a focus for an application in this manuscript. Although, we envision loading diverse diagnostic agents or removing contaminants from water via PPSU homopolymer self-assembly, we have removed discussion of these potential applications from the text.

The following statement was therefore removed:

"Our results demonstrate that sulfone-sulfone bonded networks have immense potential for applications in water purification, nanomedicine and diagnostics"

The following text was added to clarify:

“We demonstrate that sulfone-sulfone bonded networks have exceptional molecular capture capability, providing a new type of drug delivery platform that would greatly enhance the ability to encapsulate therapeutic molecules. Although there have been intensive studies on delivery via self-assembled nanocarriers in the past few decades, substantial challenges remain due in part to their low loading of water-soluble biologics and hydrophilic small molecule therapeutics. Drug loading in polymeric vesicles (i.e. polymersomes) for example are typically lower than 20%; some are even substantially less than 5%³¹.”

“PPSU is therefore a versatile platform for supramolecular chemistry that may have extensive applications in areas requiring efficient molecular encapsulation, such as biomedicine.”

2) Stability. Show stability of particles over time using a technique such as DLS, to demonstrate that the particles are stable as described.

The reviewer has made an excellent suggestion. We followed the size evolution of PPSU₂₀ vesicular nanogels over time using DLS and the results are now presented in the new Table S5. Number average diameter showed no statistically significant difference after aging for 20 days. Representative DLS plots were also shown in the new Fig. S20, giving the size distribution of PPSU₂₀ vesicular nanogels in the range of 20–100 nm. This size distribution remained stable over the 20 days without occurrence of any additional peaks to indicate nanogel aggregation. These results thus demonstrated that the nanogels were stable in water over time.

We have thus added the following statement to the text:

“PPSU₂₀ nanostructures were highly stable in water and no aggregation or precipitation were observed when aging a suspension of vesicular nanogels for 20 days at room temperature (Table S5, Fig. S20).”

3) Figure S24-25. The materials appear to be cytotoxic at higher concentrations. The non-cytotoxic concentration range should be better defined, and statistics should be performed.

We thank the reviewer for pointing out this detail. Statistics was performed for the cytotoxic study and the results are now included in Fig. S27. Although limited toxicity has been found by the MTT assay (Fig. S26, no statistically significant difference was found between the experimental materials and the PBS control group), the live/dead staining experiments (Fig. S27) did show cytotoxicity at higher concentrations, e.g., 0.5 mg/mL.

We have updated this finding in the text with the following statement:

“For their potential use in biomedical applications, we further demonstrated no cytotoxicity up to 0.25 mg/mL (Figs. S26 and S27)”.

4) Figure S21. In the measurement of fluorescent BSA adsorption onto particles surfaces, the authors followed the fluorescence of the supernatant. No decrease in fluorescence was observed. It

would be better to follow the fluorescence of the particles directly, as the decrease in supernatant fluorescence may not be significant even with protein adsorption.

Thank you very much for the suggestion of using the fluorescence of the nanogels directly. We updated the figure as a new Fig. S22a and acknowledge that the negligible FITC fluorescence of the FITC-BSA-adsorbed nanogels demonstrated weak protein adsorption. Note that FITC-BSA molecules were highly concentrated in nanogels and the fluorescence was partially quenched after encapsulation. We also included a method to quantify the protein adsorption in a new Fig. S22b. It is shown that fluorescence of FITC-BSA can be recovered by breaking nanogels in aqueous NaOH solution. Quantitative fluorescence analysis confirmed that the protein adsorption was 6.0%, while protein encapsulation was 97.8%. We provided the experimental details in Supplementary Materials and the results presented in Fig. S22 demonstrated that the high encapsulation of BSA is not due to protein adsorption.

5) Figure 4H. Why was SEM and not standard deviation used for panel H? Is the y-axis in panel H equal to $F(+)/F(-) * 100$? The statement "with a 5% significance level" is not required and confusing.

We thank the reviewer for their inquiry into the statistics reported in Figure 4H. In this experiment, an ANOVA with a post hoc multiple comparisons test was employed to determine statistically significant differences in the mean % change in fluorescence across three samples. Since ANOVA examines the equality of means (i.e. $\mu_0 = \mu_1 = \mu_2$), and the multiple comparisons test is used to determine which (if any) of the means are significantly different, we found it most appropriate to report the mean value with the uncertainty described using the standard error of the mean (s.e.m.).

The y-axis in panel H is the percentage increase in FITC fluorescence (i.e., $100 \times (F_{(+)} - F_{(-)}) / F_{(-)}$) after proteolysis, where $F_{(+)}$ is the fluorescence after protease treatment and $F_{(-)}$ refers to the fluorescence in the absence of protease treatment.

These details were added to both the figure caption and the methods section:

"The percentage increase in FITC fluorescence (i.e., $100 \times (F_{(+)} - F_{(-)}) / F_{(-)}$) after trypsin proteolysis was calculated, where $F_{(+)}$ is the fluorescence after protease treatment and $F_{(-)}$ refers to the fluorescence in the absence of protease treatment."

Furthermore, we have removed the 5% significance level statement to avoid confusion.

6) Figure 4. The reason for particle aggregation as a function of BSA concentration is not clear. Is it due to BSA-BSA interactions? Is it due to the denaturation of BSA, which exposes sticky hydrophobic domains? If so, does very high BSA concentrations prevent aggregation due to free BSA in solution? A better explanation is required.

We agree with the reviewer that it is instructive to discuss the reason for particle aggregation as a function of BSA concentration. We originally wrote the following text to describe this process:

"These results suggest that FITC-BSA is accessible on the surface of the nanogels at densities dependent upon the initial loading concentration, and one potential explanation for this is

described in the schematic of Fig. 4a. The lower concentration (0.1 mg/mL) resulted in minimal surface exposure of FITC-BSA, while the intermediate (1 mg/mL) loading concentration resulted in a partial surface exposure that allowed sharing of surface exposed protein between different nanogels to induce aggregation.”

We understand that this may not have been sufficient, so we performed additional experiments and modified the text for clarity. The particle aggregation is likely not due to BSA-BSA interactions or the denaturation of BSA, since much smaller nanogels formed when BSA was overloaded. Furthermore, BSA has a net negative charge and its adsorption to negatively charged surfaces results in minimal unfolding, with negatively charged residues and domains being oriented away from the surface (J. Phys. Chem. B 2016, 120, 10463–10468; J. Phys. Chem. B 2017, 121, 16, 3975–3986; - these references were added to the main text), which would maintain the negative surface charge of the PPSU nanogels and inhibit, not encourage aggregation. This result is supported by our zeta potential analysis in Table S6 of PPSU nanogels with and without adsorbed albumin. Additionally, overloading of BSA using a high protein concentration (10 mg/mL) prevented aggregation, likely due to solution-accessible BSA onto the surface of nanogels. When we removed free BSA from the overloaded samples, no significant changes were detected for their size by DLS, indicating that free BSA molecules have limited influence on nanogels aggregation. These results are presented as a new Fig. S24.

Further analysis is indeed warranted to investigate protein interactions at PPSU interfaces but is beyond the scope of this current manuscript. Our group works in the area of surface science with an emphasis on protein adsorption and denaturation at nano/bio interfaces (Vincent et al. bioRxiv 2020 doi: <https://doi.org/10.1101/2020.04.24.060772>; Scott et al. Biomaterials. 2007 Sep;28(27):3904-17. doi: 10.1016/j.biomaterials.2007.05.022), so we have a great deal of interest in this process. We kindly request the opportunity to complete a detailed analysis of this process for PPSU and present this data in a focused publication in the future (we are currently seeking funding for this work now).

Based on the above-mentioned data and literature, the text has been updated as follows for clarity:

“These results suggest that FITC-BSA is accessible on the surface of the nanogels at densities dependent upon the initial loading concentration, and one potential explanation for this is described in the schematic of Fig. 4a. The lower concentration (0.1 mg/mL) resulted in minimal surface exposure of FITC-BSA, while the intermediate (1 mg/mL) loading concentration resulted in a partial surface exposure during PPSU network collapse, which may have allowed sharing of protein between different nanogels to induce aggregation (Fig. 4d). Since free FITC-BSA molecules have limited influence on nanogels aggregation (Fig. S24), the saturated loading concentration (10 mg/mL) likely resulted in a high surface density of exposed FITC-BSA, which prevented aggregation between nanogels at higher BSA concentrations (Fig. 4e). Due to its net negative charge, BSA is not extensively denatured on negatively charged surfaces and has been predicted to orient with negative residues and domains exposed to the aqueous solution after adsorption^{29,30}. This is supported by our zeta potential analysis shown in Table S6 and indicates that adsorbed BSA would not induce the PPSU nanogel aggregation observed

in Fig. 4d. We employed trypsin digestion to assess payload accessibility, and ~30-35% of encapsulated FITC-BSA was accessible to cleavage for the overloaded 10 mg/mL sample (Figs. 4g-h and Fig. S25), suggesting considerable solution exposure of loaded protein.”

REVIEWERS' COMMENTS:

Reviewer #1 (Remarks to the Author):

I have reviewed this manuscript before and deemed it suitable for publication after all my concerns were appropriately addressed by the authors. I have had now the opportunity to revise this paper again (on behalf of another reviewer). The concerns raised are, in my view, completely valid and show the depth at which that reviewer evaluated the manuscript. I commend the authors that have responded to those comments with a matching level of rigor. The manuscript text was modified significantly, key data was re-analyzed, and a significant amount of new experiments (DLS, cytotoxicity, and protein quantification) were performed. I have confidence that this work is now ready and I recommend it to be published in Nature Communications. - Cecilia Leal.

Reviewer #3 (Remarks to the Author):

The authors have adequately addressed my comments with additions which improve clarity of the MS. This reviewer thanks the authors for their work and can recommend this article for publication in Nature Communications.